# A new global surface temperature reconstruction for the Last Glacial Maximum

James D Annan[1], Julia C Hargreaves[1], and Thorsten Mauritsen[2]

[1]Blue Skies Research Ltd, The Old Chapel, Albert Hill, Settle, BD24 9HE, UK
[2]Department of Meteorology, Stockholm University, Stockholm, Sweden

*Correspondence to:* jdannan@blueskiesresearch.org.uk

**Abstract.**

We present a new reconstruction of surface air temperature and sea surface temperature for the Last Glacial Maximum. The method blends model fields and sparse proxy-based point estimates through a data assimilation approach. Our reconstruction updates that of Annan and Hargreaves (2013), using the full range of GCM simulations which contributed to three generations of the PMIP database, three major compilations of gridded SST and SAT estimates from proxy data, and an improved methodology based on an Ensemble Kalman Filter. Our reconstruction has a global annual mean surface air temperature anomaly of $-4.5 \pm 0.9°C$ relative to the pre-industrial climate. This is slightly colder than the previous estimate of Annan and Hargreaves (2013), with an upwards revision on the uncertainty due to different methodological assumptions. It is, however, substantially less cold than the recent reconstruction of Tierney et al. (2020). We show that the main reason for this discrepancy is in the choice of prior. We recommend the use of the multi-model ensemble of opportunity as potentially offering a credible prior, but it is important that the range of models included in the PMIP ensembles represent the main sources of uncertainty as realistically and comprehensively as practicable if they are to be used in this way.

## 1 Introduction

There is a significant demand for reconstructions of the large spatial patterns of paleoclimatic states, in order to understand past climate changes and how these may relate to expected future changes (eg Sherwood et al., 2020). Estimating these states is far from trivial, as climate proxy data are very limited in time and space, and also contain substantial uncertainties and potentially systematic biases. Model simulations produce gridded fields, but models are imperfect and the boundary conditions used to drive them are uncertain. The aim of this work is to combine the strengths of models and data in order to produce robust and globally complete estimates of the temperature fields including credible estimates of uncertainty. While we focus here on a specific time period, the methodological issues addressed are general and we plan to apply a similar approach to reconstructions of other epochs.

Here we present a new reconstruction of the Last Glacial Maximum (LGM). We use the diverse ensemble of model simulations of past climate states which were generated by the GCMs which participated in various community model intercomparison projects, together with comprehensive data sets over land and ocean, in order to generate spatially complete and physically

coherent maps of surface air temperature (SAT) and sea surface temperature (SST) for both periods. Our approach has some similarities to the methods of Annan and Hargreaves (2013) (henceforth AH13) and Tierney et al. (2020) (henceforth TEA20) but differs in several important ways and the resulting LGM reconstruction is substantially different from both of these previous analyses.

The Last Glacial Maximum (LGM, 19-23ka BP) is the most recent period in which the global climate was broadly in a quasi-equilibrium state very different to the modern climate, and as such has been widely investigated and used both to test the ability of models to simulate the response to substantial radiative forcing, and also to estimate the equilibrium climate sensitivity (Schneider von Deimling et al., 2006; Schmidt et al., 2014; Sherwood et al., 2020). The climate system at this time was primarily characterised by a lower level of atmospheric $CO_2$ and the presence of large terrestrial ice sheets (Braconnot

et al., 2007). The spatial pattern of temperature change at the LGM is non-homogeneous, due in large part to the large ice sheets over the northern hemisphere whose state is not directly recorded in climate proxy data. This makes reconstruction of the spatial pattern of the temperature field a challenging target. A number of previous studies have produced reconstructions of temperature fields over land and/or ocean (Rind and Peteet, 1985; Schmittner et al., 2011; Annan and Hargreaves, 2013; Kurahashi-Nakamura et al., 2017; Tierney et al., 2020; Paul et al., 2021).

In AH13, we used a heuristic multi-model pattern scaling approach using multiple linear regression to generate fields for surface air temperature and sea surface temperature at the LGM. While this method successfully reproduced the large-scale features of the LGM, it was not able to fit more localised features that may exist in the data but not in the model simulations. This previous reconstruction also contained small-scale features in areas which were sparse in data, which we suspected were possibly the result of noise that was artificially generated by the reconstruction method.

The method we use here for our new analysis is a Bayesian approach based on ensemble Kalman filtering, similar to that of Tierney et al. (2020). The conceptual framework for this method is the use of a set of model simulations as a probabilistic prior estimate for the climate $\Theta$, which is then updated in the light of proxy-based observational evidence $O$ using Bayes' Theorem:

$$P(\Theta|O) = P(O|\Theta)P(\Theta)/P(O). \tag{1}$$

Here $P(\Theta|O)$ is the posterior probability distribution of the climate state conditioned on the set of observations $O$, $P(O|\Theta)$ is the likelihood function that indicates the probability of obtaining the observations given a particular climate state $\Theta$, $P(\Theta)$ is a prior distribution for the climate state, and $P(O)$ is the probability of the observations which is required as a normalising constant in the calculation of the posterior probability distribution.

    The specification of the prior distribution $P(\Theta)$ for our LGM reconstruction is based on a larger set of model simulations

than were available to AH13 and is outlined in Section 2. A new analysis and compilation of proxy data has been presented by Tierney et al. (2020) and in Section 3 we discus these and other proxy data that we use which, together with their uncertainties, gives rise to the likelihood $P(O|\Theta)$. In Section 4 we describe the calculation of our posterior estimate, i.e. the temperature reconstruction, using an ensemble Kalman filter data assimilation method, and also present a number of sensitivity tests and

validation exercises to investigate the robustness of the results. We discuss how and why our results differ from previous work in Section 6.

## 2 Models and prior

Our prior is based on an 'ensemble of opportunity' consisting of the set of 31 LGM simulations generated by a range of structurally distinct climate models which contributed to several Paleoclimate Model Intercomparison Projects: PMIP2 (Braconnot et al., 2007), PMIP3 (Braconnot et al., 2012) and PMIP4 (Kageyama et al., 2018, 2021), which we refer to collectively as PMIP. The PMIP experiments were initially separate from, but more recently an official component of, the Coupled Model Intercomparison Project (CMIP) suite of modelling experiments, which means that the experimental protocols are publicly available and the experiments are open to all modelling groups with sufficient interest and resources to take part. The simulations consist of an ad hoc subset of all climate models. Due to the limited availability of model outputs, climatologies are calculated as 30-year averages for PMIP2 and PMIP3, and 50-year averages for PMIP4.

### 2.1 Model selection

The model simulations that are available to us are listed in Table 1. After regridding the available outputs to a regular 2 degree grid for surface air temperature and 5 degree grid for surface ocean temperature, it was clear that three of the models (which are indicated by the letter 'G' in the table) had unusual ocean boundaries such that substantial coastal regions were masked out of the ocean. This would significantly reduce the number of climate proxy data points that we could use in our reconstruction, and therefore we we did not consider these three models further, leaving us with 28 models for further consideration.

We also recognise that this meta-ensemble contains several near-duplicate models which share a common heritage (Knutti et al., 2013), and therefore our next step is to perform a filtering to reduce this set of simulations to a set which we can more reasonably regard as independent (Annan and Hargreaves, 2017). Based on the principles presented in Annan and Hargreaves (2017), we should base our decisions over which models to exclude on our a priori belief in model similarity, perhaps best considered as exchangeability in the Bayesian statistical sense. It would not be appropriate to automatically filter models based on the similarity of their outputs, as in principle this would penalise the best models, which are necessarily quite similar to each other as they will also by definition be most similar to reality. Our a priori expectation is that the various models from a single centre will be largely exchangeable (e.g. the numerous Hadley Centre models, the MIROC family, the CESM/CCSM models, versions of IPSL models, and two versions for each of LOVECLIM, MPI, AWIESM, FGOALS and CNRM), and also models that we know to be partially developed from a common source, specifically ECBILTCLIO being closely related to the LOVECLIM models and ECHAM53-MPIOM127 being an early version of the MPI model.

However, we have limited direct knowledge about the design and structure of the wide range of models available to us and therefore we chose to augment and validate this prior judgement with an a posteriori measure of pairwise model similarity in terms of their LGM anomaly fields, as measured by both pointwise RMS difference and also pattern correlation. For the

| Experiment | Model Name | Issues |
|---|---|---|
| PMIP2 | ECBILTCLIO | D |
| PMIP2 | CCSM_ncea | |
| PMIP2 | CNRM-CM33 | |
| PMIP2 | FGOALS-1.0g | |
| PMIP2 | HadCM3M3 | |
| PMIP2 | IPSL-CM4-V1-MR | |
| PMIP2 | MIROC3.2.2 | D |
| PMIP2 | HadCM3M3 (V) | D |
| PMIP2 | ECHAM53-MPIOM127 | |
| PMIP3 | CCSM4 | G |
| PMIP3 | CNRM-CM5 | G |
| PMIP3 | FGOALS-g2 | |
| PMIP3 | GISS-E2-R | |
| PMIP3 | IPSL-CM5A-LR | D |
| PMIP3 | MIROC-ESM | G |
| PMIP3 | MPI-ESM-P | |
| PMIP3 | MRI-CGCM3 | |
| PMIP4 | AWIESM1 | D |
| PMIP4 | AWIESM2 | D |
| PMIP4 | CCSM4 | |
| PMIP4 | CESM1-2 | |
| PMIP4 | CESM2-1 | O |
| PMIP4 | HadCM3-GLAC1D | |
| PMIP4 | HadCM3-ICE6GC | D |
| PMIP4 | HadCM3-PMIP3 | |
| PMIP4 | INM-CM4-8 | |
| PMIP4 | IPSL-CM5-A2 | |
| PMIP4 | MIROC-ES2L | |
| PMIP4 | MPI-ESM1-2 | |
| PMIP4 | iLOVECLIM1-1-1-GLAC-1D | D |
| PMIP4 | iLOVECLIM1-1-1-ICE-6G-C | |

**Table 1.** Models available for the LGM reconstruction. G indicates removal for gridding problems. D indicates removal for duplication/similarity, O indicates removal as an outlier. PMIP2 data were sourced from PMIP modellers (1998), PMIP3 data from PMIP modellers (2017), and PMIP4 data from J.Y. Peterschmitt, and PMIP modellers (2021)

most part, these measures merely confirmed what we already believed to be the case. However, in one case (specifically, the AWIESM models being similar to the ECHAM and MPI models), this check alerted us to a model relationship that we had not been previously aware of but could identify in retrospect from literature, and in another case (the CCSM and the CESM variants), it suggested that changes between model versions had been more substantial than anticipated.

Based on these analyses, we therefore concluded that the following groups of models were unusually similar to each other and required thinning: (ECBILTCLIO, iLOVECLIM1-1-1-GLAC-1D, iLOVECLIM1-1-1-ICE-6G-C), (HadCM3M3, HadCM3M3(V), HadCM3-GLAC1D, HadCM3-ICE6GC, HadCM3-PMIP3), (IPSL-CM4-V1-MR, IPSL-CM5A-LR, IPSL-CM5-A2) (MIROC3.2.2, MIROC-ES2L), (ECHAM53-MPIOM127, MPI-ESM-P, AWIESM1, AWIESM2, MPI-ESM1-2). As the large number of models based on versions of CCSM/CESM all appear to differ substantially from each other, we retain all of them at this point.

From the group of LOVECLIM models, we remove ECBILTCLIO and iLOVECLIM1-1-1-GLAC-1D, keeping iLOVECLIM1-1-1-ICE-6G-C since it is most representative of the mean of this group of models. From the IPSL group we drop the intermediate IPSL-CM5A-LR version, keeping both the older IPSL-CM4-V1-MR and the highest resolution IPSL-CM5-A2 model which are more substantially different from each other. We keep the more recent MIROC-ES2L and remove the older MIROC3.2.2 on the basis that newer models are likely to outperform older ones. We remove HadCM3m3(V) and HadCM3-ICE6GC, retaining the three other variants in this group which again differ quite significantly. Both of AWIESM models are very similar to each other and also to the MPI-ESM1-2 from which they are derived, so we retain the latter only.

This thinning of the ensemble increases the minimum pairwise area-weighted RMS difference between the modelled LGM anomaly fields from $1°C$ to $2.3°C$, removing at least one member from each of the closest 10 pairs of models in the original ensemble. Sensitivity tests which we present in Section 5.2 show that in fact this thinning process has little effect on our overall results.

Our assimilation technique (presented in Section 4) relies on the assumption of a Gaussian prior, but even after this removal of near-duplicates, the ensemble of global mean temperature anomalies calculated by the simulations still has a striking outlier in the form of CESM2-1 (Zhu et al., 2021) such that the ensemble of global mean temperature anomalies fails a Shapiro-Wilks normality test at the $p < 0.001$ level. Therefore, we remove this model, at which point the Shapiro-Wilks test on global mean anomaly is no longer rejected, with a $p \simeq 0.46$. The final ensemble that we use for our reconstruction contains 19 simulations, which compares favourably with the 9 models that were used by AH13.

Figure 1 shows histograms of the global mean surface air temperature of the 28 models that we analysed, with the blue colour representing the 19 models that we take forward into the data assimilation, and the red colour indicating the 9 models that we omitted, one (CESM2-1) as an obvious outlier and the other 8 due to the dependencies described above.

The original 28-member meta-ensemble, before we perform the thinning process, has a mean globally-averaged temperature anomaly of $-5.0°C$ with a standard deviation of $1.6°C$. After the removal of duplicates and the CESM2-1 outlier, the global mean surface air temperature anomaly of the remaining 19-member ensemble is $-4.9 \pm 1.1°C$.

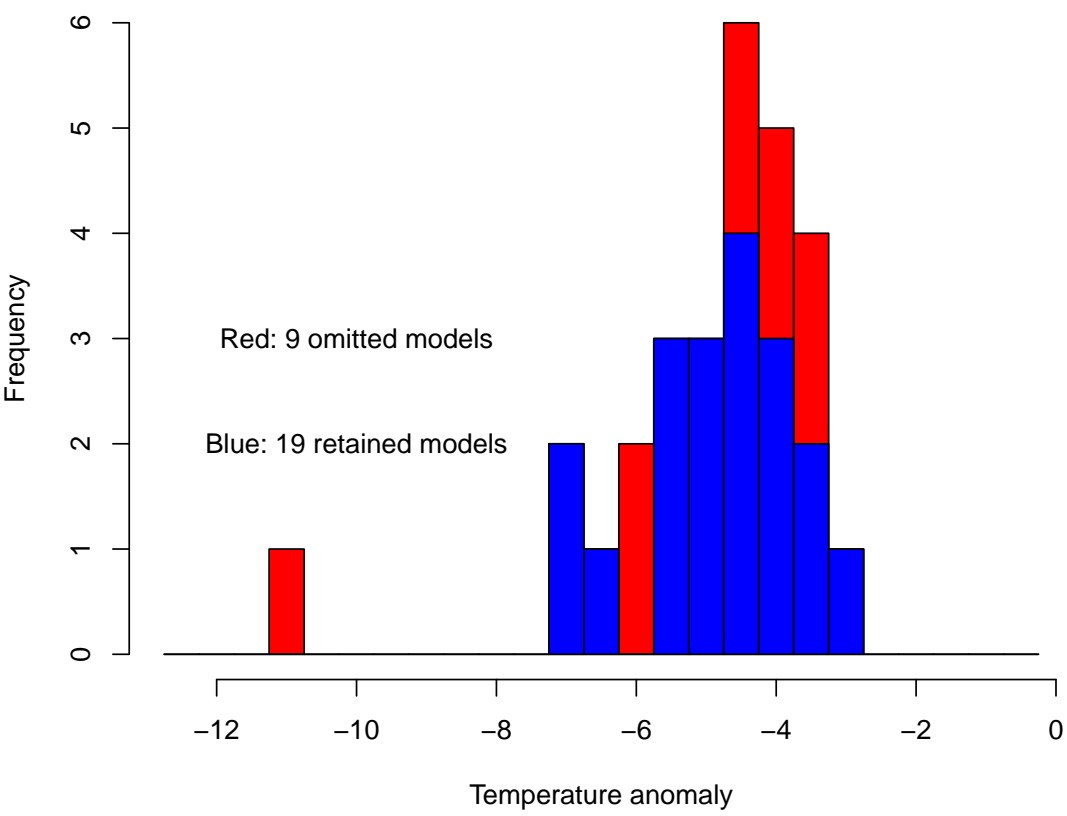

**Figure 1.** Histogram of global mean surface air temperature of PMIP models. Blue: retained models. Red: omitted models.

## 2.2 Debiasing the prior

Even after this thinning process, we are not yet confident that the ensemble can be considered as a credible prior, as the inclusion of models in PMIP experiments themselves is rather arbitrary due to contingencies such as the motivation and interests of research staff and the availability of sufficient resources, and the total number of simulations is small. While similar 'ensembles of opportunity' have frequently been used as a representation of uncertainty, there is increasing recognition that this is a somewhat risky choice to make (Tebaldi and Knutti, 2007; Knutti, 2010; Tokarska et al., 2020). It is quite plausible that the entire set of results could have a bias that exceeds its spread (and thus excludes reality) due to limitations in both model structure and experimental design. For example, it could be the case that the models all have a climate sensitivity that is either too high, or too low, when compared to the truth. Furthermore, the boundary conditions defined for the PMIP experimental protocols could be biased in a number of ways, with resulting global or regional biases in the ensemble of model fields.

While we do of course use observations to further constrain the paleoclimate reconstructions, the data are sufficiently sparse and uncertain that we anticipate the potential for significant sensitivity to the prior, which we show to be the case in Section 5.2. Therefore, in order to address these issues, before starting the data assimilation procedure, we perform a translation operation on the 19-member ensemble in order to ensure that it is approximately centred on the data. We do this by firstly performing a pattern scaling analysis similar to that presented by AH13 in order to generate a field that is close to the data points, and then centre the ensemble on this field. The purpose of this operation is to ensure that the ensemble mean is a reasonable initial estimate of the climate state, without having large biases.

The pattern scaling algorithm to calculate the new ensemble mean follows the approach described in AH13, but instead of using the full set of model anomaly fields as predictors, we only use the first 4 EOFs of this ensemble in order both to reduce noise in the fitted field and also to reduce the number of predictor variables. That is, we identify 4 scaling factors $\alpha_i$, $i = 1 \ldots, 4$ to apply to the first 4 uncentred EOFs $E_i$, $i = 1, \ldots, 4$ so as to minimise the RMS difference between the data points and the pattern defined by $\sum_i \alpha_i E_i$.

The first 4 EOFs represent large scale patterns such as the overall cooling pattern, and latitudinal and land-sea contrasts, though they have no direct physical interpretation. Due to the uncentred approach, the first EOF mode is close to the ensemble mean and represents 92% of the total variance, with the next three EOFs only representing between 2.6 and 0.67% of the total variance. However, collectively this amounts to 60% of the remaining variance after the first EOF is removed.

The result of this pattern scaling is then used as the mean of the translated ensemble. This allows us to fit the largest scale patterns but only uses 4 degrees of freedom in this initial fit to the data set, which contains 405 data points. In our sensitivity tests (Section 5.2), we consider the effect of using a different number of EOFs in this step.

The ensemble translation is performed by applying an identical linear translation operation on each ensemble member. That is, we replace each model field $m_i$ with the field

$$m_i' = m_i + \mathbf{q} - \mathbf{m} \tag{2}$$

where $\mathbf{m}$ is the initial mean of the ensemble, and $\mathbf{q}$ is the field obtained from the pattern scaling algorithm. In performing this translation of the ensemble, we retain the covariance structure of the ensemble by translating each member in the same manner.

The initial ensemble mean (after the thinning process described earlier) has a mean bias relative to the data points of $0.5°$ C, with a pointwise RMS error of $2.8°$ C. The field obtained by the pattern scaling algorithm has a mean bias of $0.2°C$ with the RMS error marginally reduced to $2.7°C$. Thus, the pattern scaling is not fitting the individual data points closely, but the mean bias (which was already modest) is reduced to a very low level by this translation.

After translation, the global temperature anomaly of the ensemble mean is reduced to $-4.4°C$, with the spread by construction remaining unchanged. Sensitivity tests show that this translation in the present application has very little impact on the final results. Although this result suggests that this step could have been omitted, we show in Section 5.2 that the translation operation would be crucial if the initial ensemble had in fact been strongly biased. Thus we consider it a useful step to take in general.

Figure 2 shows the mean and uncertainty (i.e. pointwise standard deviation) of the ensemble after the pre-processing that was previously described, along with the data points that we are using. We describe the data in the following section.

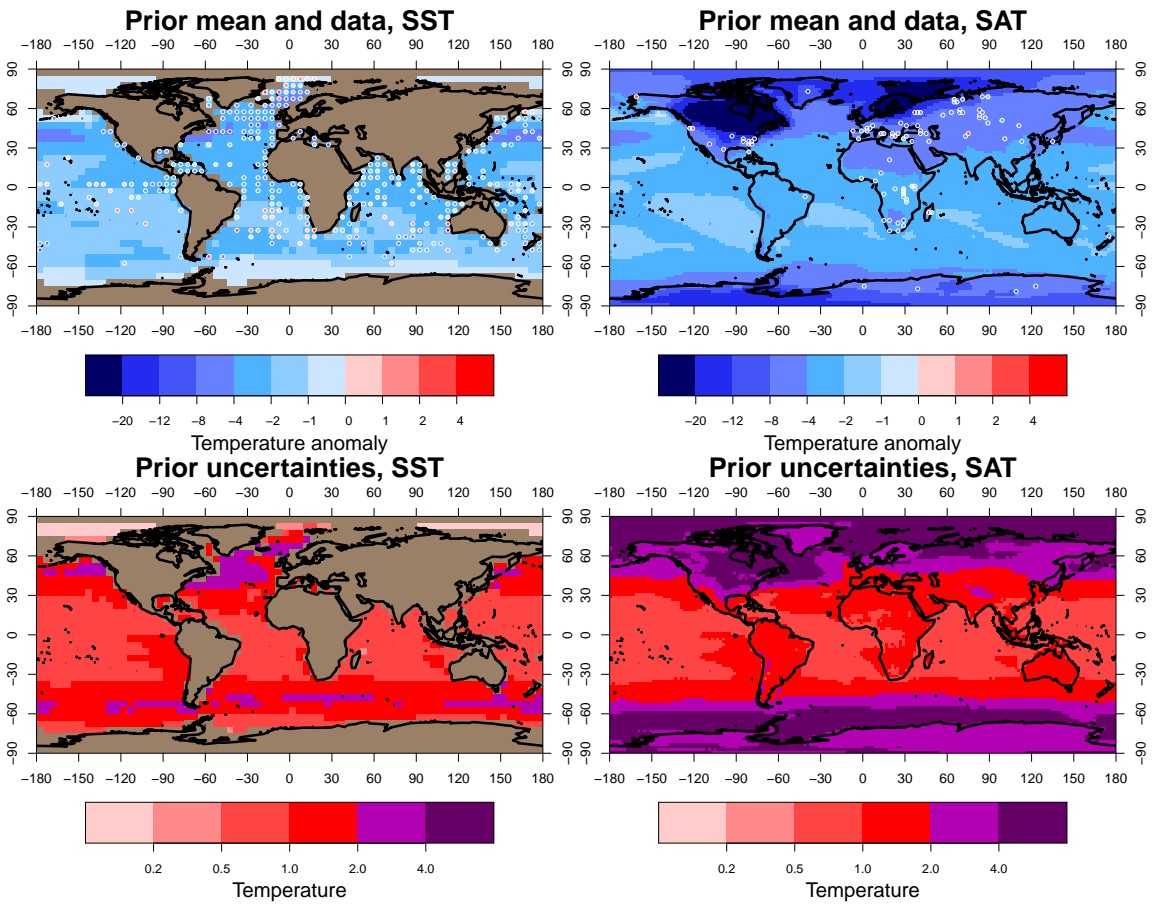

**Figure 2.** The prior: translated ensemble mean, and data. Lower plots, uncertainties. SST and SAT, left and right

## 3 Data, uncertainties, and likelihood

Our LGM reconstruction relies primarily on three syntheses of data: the sea surface temperature (SST) analysis of the MARGO project (MARGO Project Members, 2009), the surface air temperature (SAT) compilation of Bartlein et al. (2011) and the more recent SST compilation of TEA20. The pollen-based land surface air temperature data of Bartlein et al. (2011) was augmented with ice core data by Schmittner et al. (2011) and we use this larger data set in our analyses. The MARGO and TEA20 data are available both as absolute values and as anomalies relative to an estimate of pre-industrial climate. In the case of MARGO, these anomalies are calculated relative to the World Ocean Atlas database (Conkright et al., 1998), but in the case of TEA20, they are calculated relative to their own 'core-top' (representative of late Holocene) data. Thus, the two sets of anomalies

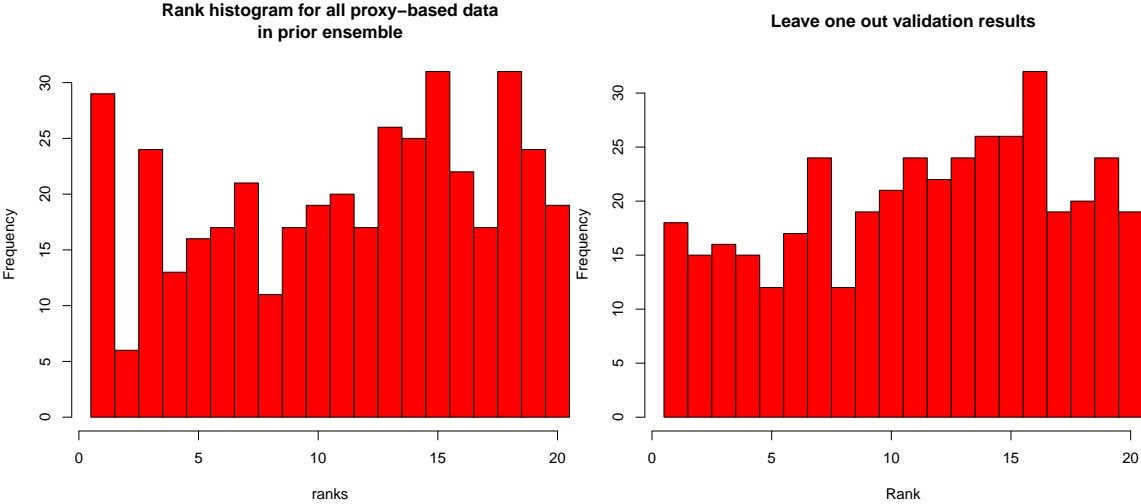

**Figure 3.** Left: Rank Histogram of proxy data in prior ensemble, showing the position of each data point in the ordered set of model values. Right: Rank Histogram of withheld proxy data in posterior ensemble in leave-one-out validation tests.

are not calculated on a consistent basis, and moreover, the TEA20 anomaly data are only available at the subset of points where LGM and core-top data are both present. We use annually averaged temperature throughout. While many proxies are determined more directly by seasonal temperatures, Annan and Hargreaves (2015) found that performing their analysis on a seasonal basis (with appropriately-calibrated proxy-based temperature estimates) made little difference to the overall result.

While our method uses temperature anomalies, the absolute temperatures from TEA20 and MARGO are in fact in marginally closer agreement to each other than the anomaly data are, at the grid points where both data sets exist. We therefore take the absolute data from both sources to use in our analysis, and calculate a new set of anomalies on a regular 5° grid relative to a consistent baseline for both data sets. The ERSSTv5 data set (Huang et al., 2017) provides an updated and improved reanalysis of SST over the pre-industrial period, which should more closely represent a true pre-industrial state than the WOA database

that MARGO previously used. We therefore use the ERSSTv5 data averaged over 1851-1880 as our pre-industrial baseline for both LGM SST data sets. The TEA20 data contains 957 data points but these only span 282 of our 5°grid squares with many of these grid squares containing multiple data points. In these cases we take the simple mean due to our treatment of uncertainties which is discussed in more detail below. Over land, we use the data set from Bartlein et al. (2011) in anomaly form on a 2°grid, with the addition of a few more data points such as the ice core data provided by Schmittner et al. (2011).

When limited to locations for which all climate models have SST outputs after regridding to the same 5° grid, the MARGO data set contains 258 usable data points, the TEA20 data set has 204 remaining points, with the number of overlapping locations being 152 points. Rather than attempting a blend of MARGO and TEA20 data where they coincide, we choose to simply use the TEA20 data where both are available, on the grounds that it benefits from an additional decade of research and so could be considered more reliable. Thus, the 204 points from TEA20 are augmented by an additional 106 points from the MARGO

data set, making 310 SST points in total. We have 95 SAT points on a 2° grid from the augmented Bartlein et al. (2011) data set, so the total data set contains 405 points. The combined data set that we use in our reconstruction is shown by the coloured dots on Figure 2.

As a check of the calibration of the ensemble we present in Figure 3 the rank histogram of the data that we are using in the
ensemble. The rank of an observation is its position in the ordered set of model predictions for the same observation, i.e. a rank of $i + 1$ indicates that $i$ models in the ensemble predict a smaller value for this observation, and the remaining $n - i$ predict a larger value. The model predictions here must account for observational error, which we discuss below. If the ensemble is biased high then the rank histogram will tend to be skewed left, and vice versa, and if the ensemble spread is too small and therefore insufficiently dispersed, then the rank histogram will tend to be u-shaped with many data points having rank 1 or
$n + 1$. Any of these results would have cast doubt on the use of the ensemble as a credible prior for the reconstruction. In our case, the rank histogram is roughly flat (albeit with substantial sampling noise due to the limited number both of ensemble members and also data points), indicating that the ensemble spread is quite well calibrated. It must be recognised however that examining the uniformity of the rank histogram is a rather limited test (Hamill, 2001) and we only present it here as an indication that the prior is not obviously flawed, rather than to claim that the prior is ideal.

We now consider the uncertainties of the data. The MARGO data set has an assessment of quality which is widely interpreted as an uncertainty estimate, with the RMS value across the data points being 2.0°C, but having significant spread across the data with some values being lower than 1°C and others greater than 3°C. The Bartlein et al. (2011) data were also provided with uncertainty estimates and these have an RMS value of 1.5°C, again with a large spread. In AH13, however, we argued that the true errors of these data points were unlikely to be accurately represented by these uncertainty estimates, with (for example)
the most uncertain data points agreeing with the model mean rather more closely than their stated uncertainties should have allowed. For our previous analysis, we therefore adopted a simple and robust approach of assuming a uniform uncertainty estimate across all of the data points, and we maintain this approach here. The TEA20 data points are also provided with individual uncertainties, but in this case the large majority of them are already close to 2°C. The RMS difference between the TEA20 and MARGO data points is around 2.5°C at locations where they coincide, which seems reasonably consistent with
these values. We also find our posterior fit (presented in the next section) has a residual RMS difference with the data of around 2°C, and a leave-one-out analysis also supports such a value. We therefore maintain our previous approach by using a uniform value of 2°C for all data in our main analysis, and explore a number of sensitivity analyses in Section 5.3.

## 4   Data assimilation using an ensemble Kalman Filter

As outlined in Section 1, we use a data assimilation method to incorporate our proxy data into the prior estimate defined by the
ensemble of model simulations, and so produce our posterior. We are effectively using Bayes' Theorem (Equation 1) to update the prior according to the likelihood. With the assumption of Gaussian uncertainties, the likelihood of a single observation $o$ is given by the Gaussian density evaluated at the appropriate value, i.e. $\frac{1}{\sigma\sqrt{2\pi}}e^{-\frac{1}{2}\left(\frac{o-\mu}{\sigma}\right)^2}$ where $\mu$ is the corresponding climatic variable derived from the climate state (i.e. the temperature anomaly of the appropriate gridpoint in our analyses) and $\sigma$ is the

observational uncertainty. For our application in which all uncertainties are assumed independent, the joint likelihood of all observations $P(O|\Theta)$ is simply the product of their individual likelihoods. While this function is not directly evaluated in our calculation of the posterior, it underpins the formulation of the problem.

The method we use for the analysis is an Ensemble Kalman Filtering algorithm (Evensen, 2003) which uses an ensemble to approximately solve the Kalman Equations. The Kalman Equations (Kalman, 1960) provide the optimal recursive estimate for updating a linear system with Gaussian uncertainties in light of uncertain observations. The estimate can be presented through its mean $\mathbf{m}$ and covariance $P$, which are updated from their prior (or 'forecast') values $\mathbf{m}^f$ and $P^f$ to their posterior ('analysis') values $\mathbf{m}^a$ and $P^a$ by the equations

$$\mathbf{m}^{\mathrm{a}} = \mathbf{m}^{\mathrm{f}} + K\left(\mathbf{o} - H\mathbf{m}^{\mathrm{f}}\right)$$

$$P^{\mathrm{a}} = (I - KH)P^{\mathrm{f}}$$

where $\mathbf{o}$ is the vector of observed values, $H$ is the observation matrix that maps model variables to observations and $K$ is the Kalman gain matrix defined by

$$K = P^{\mathrm{f}}H^{\mathrm{T}}\left(HP^{\mathrm{f}}H^{\mathrm{T}} + R\right)^{-1}$$

where $R$ is the covariance matrix of observational uncertainties.

In the Ensemble Kalman Filter, we update each member of the prior ensemble $m_i^f$ (where $i$ indexes the ensemble members) individually using perturbed observations (Burgers et al., 1998):

$$m_i^{\mathrm{a}} = m_i^{\mathrm{f}} + K\left(\mathbf{o} - Hm_i^{\mathrm{f}} + \epsilon_i\right)$$

where $\epsilon_i$ is a random sample from the distribution of observational uncertainties. The updated ensemble members then sample the posterior distribution defined by the Kalman Equations. Any desired statistics from the posterior distribution can be calculated directly from the ensemble members themselves, for example the mean and standard deviation of global or regional temperature averages.

We use a localised Ensemble Kalman Filter algorithm, using the localisation function of Gaspari and Cohn (1999). The main purpose of localisation is to reduce the risk of spurious long-distance correlations which have no physical basis but which may arise simply from sampling variability due to the small ensemble size. The method uses the physically meaningful covariances described by the ensemble, but tapers these over distance according to a quasi-Gaussian function that decays to zero at double the characteristic length scale of 2500km. The method has commonalities with the simpler Optimal Interpolation (OI) approach in which an invariant Gaussian covariance function would be used for the update. However in contrast to OI, the Gaussian covariance function described by the localisation function is modified by the empirical covariances defined by the

| Region | Mean ± Standard Deviation (°C) |
|---|---|
| Global SAT | −4.5 ± 0.9 |
| Tropical SAT | −2.6 ± 0.6 |
| Global SST | −2.2 ± 0.5 |
| Tropical SST | −2.1 ± 0.5 |

**Table 2.** Summary of temperature reconstruction. Area-weighted average of reconstruction of SAT and SST, global and tropical (30°S – 30°N) values.

ensemble, thus allowing the representation of physically realistic and spatially varying covariances due to e.g. dynamics and topography. The choice of length scale to use is somewhat uncertain, and discussed further in Section 5.2.

We update SAT and SST simultaneously, using the same length scale for both data sets, so as to ensure physically consistent SAT and SST fields. Thus, the SAT data help to constrain SST results and vice-versa, maximising the spatial coverage of the sparse data. Since we have already re-positioned the ensemble mean closer to the data by means of the EOF pattern scaling, our approach does formally amount to an over-use of the data, but without this step we cannot be confident that we have an adequately trustworthy and unbiased model-based prior. Leave-one-out validation presented in Section 5.1 shows that this process, which only involves 4 scaling parameters, is not over-fitting the data to any detectable degree.

## 5 Results, Validation and sensitivity analyses

### 5.1 Results and validation

We present our reconstruction in Figure 4. Global and tropical (30°S – 30°N) values for SAT and SST are provided in Table 2. The global mean SAT anomaly under our baseline assumptions for the analysis is −4.5 ± 0.9°C at 1 standard deviation. The large difference between the global values for SAT and SST is due to strong air cooling over terrestrial ice sheets and also sea ice, which insulates the sea surface from the air above it. This latter factor also explains the lack of polar amplification in SST cooling, reflected in the similarity between these global and tropical averages.

The residual RMS difference between the prior mean and the data is 2.8°C, and for the posterior this reduces to 2.2°C, which seems reasonable in view of our estimation for data errors of 2°C. The correlation between posterior mean and data is reasonably high at 0.80. These values represent an improvement over the results of AH13 who reported a residual RMS difference of around 2.4° and a correlation of 0.73, albeit differences between the data sets used in the two analyses weakens the value of such a direct comparison.

As our primary validation of the analysis, we performed a full set of leave-one-out cross-validation analyses, removing each data point in turn and using it as an independent test of the method when the remaining 404 points are used in the reconstruction.

Leaving out a data point in this way enables us to check that the analysis is providing a genuinely improved reconstruction and not merely an elaborate but untrustworthy curve-fitting procedure to the data points that were used.

In each analysis, we start by performing the debiasing step based on the initial reduced ensemble of 19 simulations. The RMS value of the differences between each data point and the corresponding ensemble mean prediction at that grid point before we perform the recentering is 2.8°C, and this reduces marginally when the recentering is performed, with the RMS value (when averaged over all 405 experiments) remaining at 2.8°C to two significant figures. After the data assimilation is performed, the RMS difference between the withheld data points and the posterior mean is barely different from the posterior RMS fit to the data that were used, although it does round up from the value of 2.2°C for residuals of the fitted data, to 2.3°C to the withheld data points, when rounded to two significant figures. This gives us confidence that the reconstruction method is working well and that the pointwise accuracy of the posterior reconstruction is of this magnitude, contingent of course on the data being valid. An inevitable limitation of leave-one-out analyses such as this is that any large unrecognised bias across the entirety of the data will not be discovered, but the use of diverse data sources from a range of different researchers should minimise this risk. A rank histogram of the withheld data points, when each point is assessed versus the predictive distribution of the posterior based on assimilating the remaining 404 data points, is also shown in Figure 3. Again, this is reasonably flat, albeit with substantial sampling uncertainty.

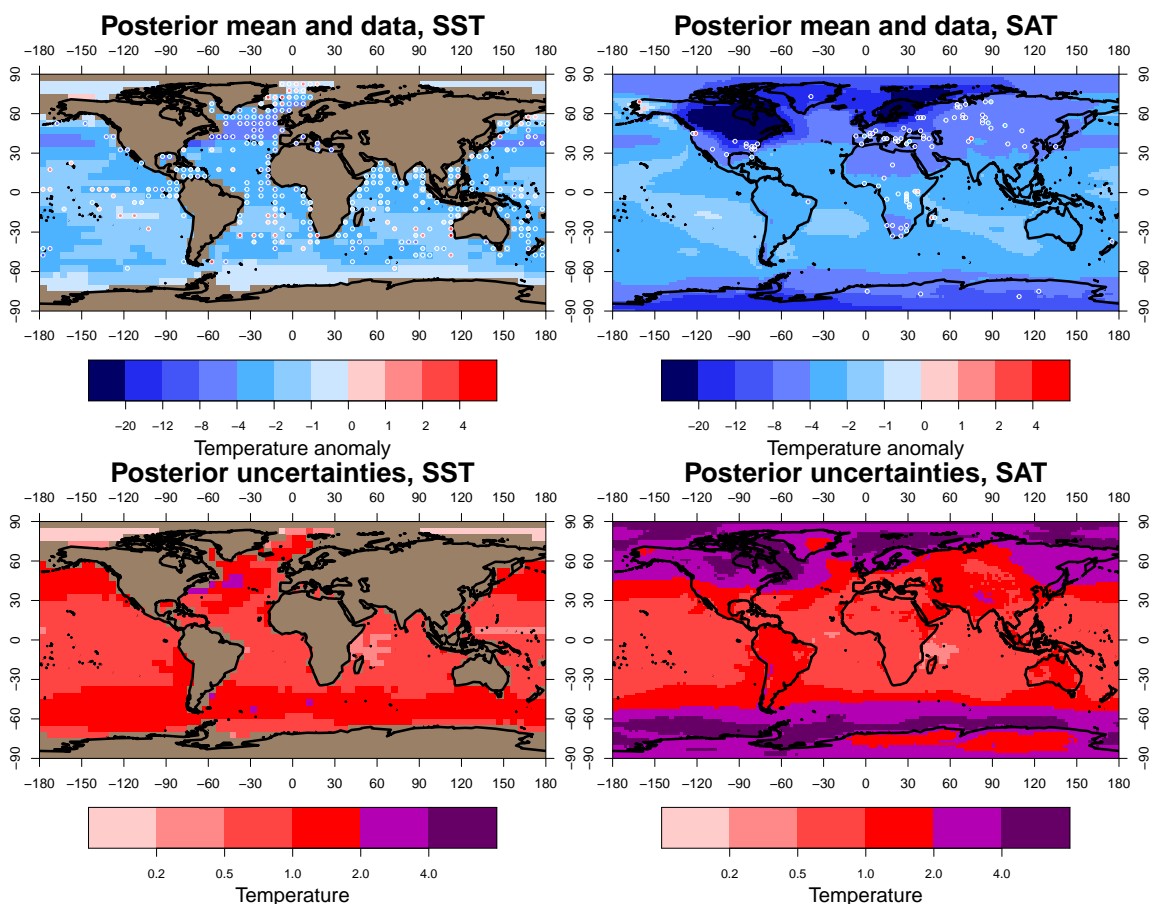

**Figure 4.** Posterior. Uncertainties in lower plots, SST left, SAT right

## 5.2 Sensitivity tests — method

In order to check the robustness of our analysis, which makes use of a number of subjective assumptions, we have performed a large number of sensitivity tests, both regarding the details of our method, and regarding the input data.

Firstly, we consider our treatment of the model prior. Our first step, as described in Section 2, was to thin the models from the original 28 to a sample of 19 that we could reasonably regard as independent. If instead we were to retain all of the 28 models and otherwise pursue the same analysis from this point, the mean of the resulting posterior temperature anomaly would be unchanged at $-4.5°$C, but the spread would be increased to $1.2°$C at one standard deviation, due to the presence of the outlier CESM2-1 model.

Returning to our preferred choice of 19 models, if we do not include the step where we translate the ensemble mean via our pattern-scaling approach, then the result becomes $-4.8 \pm 0.9°$C, slightly cooler than our baseline value. The reason for the

additional cooling here is simply that the prior ensemble mean is cooler, at $-4.9°C$ when the translation is omitted, and this initially colder state influences the result after the Bayesian updating. Therefore, this pre-processing of the ensemble has only a modest influence on the results, due to the fortuitous fact that the set of models available happened to not be too strongly biased. However, there is an improved fit to the data when the translation is applied, and there is no sign of over-fitting through this process, so we consider it warranted. If the initial ensemble had happened to be more strongly biased, it would have been a very important step as we shall now show.

We illustrate this issue by creating two alternative model ensembles, scaling the 19 model anomaly fields by a factor of 2 (respectively, 0.5) in order to consider what would have happened in the case where the prior ensemble substantially overestimates (underestimates) the true climate change at the LGM. In the case where the anomaly fields are doubled (respectively, halved), the 19-member prior ensemble has a mean temperature anomaly of $-9.7\pm2.2°C$ ($-2.4\pm0.6°C$), and the resulting posterior in the case where we implement our translation procedure becomes $-4.4\pm1.6°C$ ($-4.5\pm0.5°C$). If however we do not perform this translation of the prior mean, the posterior would instead be $-8.1\pm1.6°C$ ($-2.8\pm0.5°C$). Thus, the posterior mean is virtually unchanged in these experiments when we perform the translation, but when using the model ensemble directly as a prior, the posterior is highly dependent on the prior. In these two examples, the bias in the posterior mean is roughly 70–80% of the bias in the prior. This extremely strong dependence on the prior is a direct consequence of having limited and uncertain data.

The poor fitness of these artificial priors can also be detected by the rank histograms of the data for each ensemble. The rank histograms shown in Figure 5 are substantially skewed for both of these sensitivity tests in the case where the translation was not performed, and even in the case where we did translate the ensembles, their spread is still shown to be somewhat too broad (narrow) through the domed (u-shaped) rank histogram of the data in the translated ensemble. Thus, we conclude that while the translation operation had limited effect in the present application, it remains a valuable safeguard against the possibility that the models are significantly biased. It does not, however, remedy the situation where the ensemble is too narrow or broad.

The use of 4 EOFs in the debiasing is another place where a different choice could have been made. The main reason for using a limited number of EOFs, rather than all model fields as was done by AH13, is to reduce the presence of noise and likelihood of overfitting in the pattern. However the use of too few EOFs would limit the ability to reduce regional biases such as the degree of polar amplification and land-sea contrast which are strongly represented in the first few EOF patterns. While our choice of 4 EOFs remains a largely subjective one, using a different number of EOFs only changes the final global mean temperature estimate by up to $0.2°C$.

The data assimilation algorithm itself is conventional, but the length scale to use for the localisation is uncertain. While there has been research into the appropriate choice of length scale for application to numerical weather prediction (e.g. Kirchgessner et al., 2014), the conclusions are situation-specific. A length scale of hundreds of kilometers is typically used in application to numerical weather prediction, but it is reasonable to expect that a greater length scale is appropriate in reconstructing paleoclimates as we are considering the climate response to large-scale forcing rather than synoptic-scale variability, and also because the data are more sparse and therefore extrapolating in data-void areas is a greater concern. The small ensemble size also increases the risk of spurious long-range correlations arising from sampling error, and we would like to minimise the

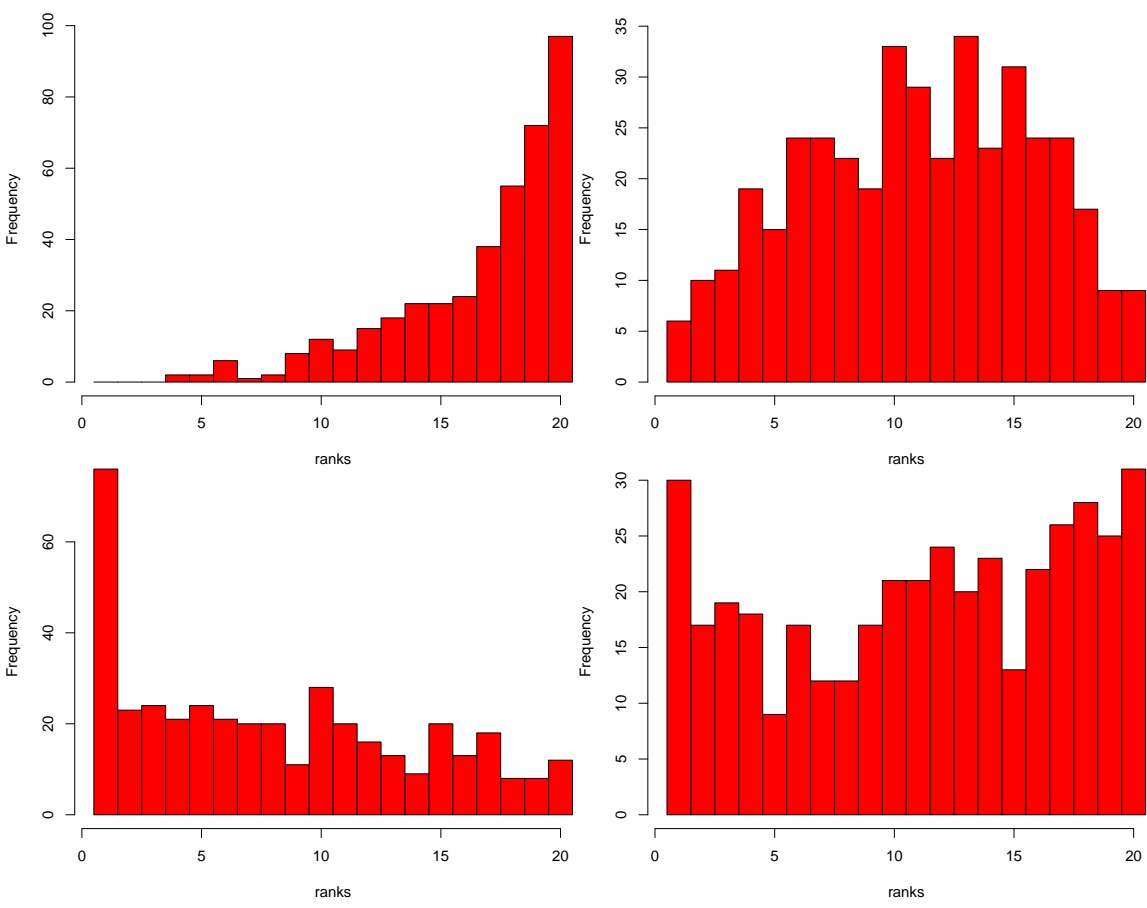

**Figure 5.** Rank histograms of data for 4 ensembles. Top left: biased ensemble with 2x scaling. Top right: biased ensemble with 2x scaling, after recentering. Bottom left: biased ensemble with 0.5x scaling. Bottom right: biased ensemble with 0.5x scaling, after recentering.

effect of these. We investigated the use of both longer and shorter length scales compared to our baseline value of 2500km. If we use a larger length scale of 10,000km, then the resulting posterior narrows to $-4.6 \pm 0.5°$C. The reason for this reduction in uncertainty is simply that each grid point is constrained by a larger number of data points. In fact the uncertainty is close to that obtained in our previous reconstruction AH13, in which there was no localisation, meaning that all data points had global

5  influence. The fit to the data, however, is slightly worsened with this longer length scale. For a smaller length scale of 1000km, the uncertainty increases slightly due to each data point having less influence, with the posterior being $-4.5 \pm 1.0°$C, barely changed from the prior. Decreasing the length scale further generates a noticeably noisy pattern of 'bulls-eye' artefacts around isolated data points without materially changing the global temperature estimate. These tests suggest that the localisation is likely to be the primary driver of the increased uncertainty in this reconstruction compared to that of AH13, and a corollary

10  of this is that we now consider our previous analysis to have been overconfident. Our preferred length scale is shorter than

that used by Tierney et al. (2020), which is primarily a consequence of ensemble size being smaller than theirs and thus more susceptible to spurious long-range correlations due to sampling uncertainties alone.

## 5.3   Sensitivity tests — data

Partly as sensitivity tests for our results, and also in order to compare with previous work, we have tested the effect of using different subsets of the data. AH13 used the MARGO Project Members (2009) and Bartlein et al. (2011) data and obtained a result of $-4.0\pm0.4°$C at one standard deviation. If we use these data sets with our new method and 19-member prior ensemble, our result becomes slightly cooler than this, with a posterior mean of $-4.4°$C, and the uncertainty is markedly higher at $0.9°$C. The slightly cooler mean relative to our previous analysis is due to slightly different model patterns extrapolating into the data void regions. When we switch from the MARGO SST data to that of Tierney et al. (2020), while still using the land data from Bartlein et al. (2011), the result cools a little further to $-4.6\pm0.9°$C. Thus, the TEA data generate a slightly colder result than the MARGO data, consistent with them being colder at their co-located by about $0.3°$C. Using the two SST data sources in their alternative form as anomalies, with the inconsistent baselines and slightly different spatial coverage that this implies, has a near-zero effect on all of our analyses and does not affect this conclusion.

If we use the TEA data alone for the analysis, with no land data at all, the resulting anomaly is slightly warmer than our main result, at $-4.4\pm1.0°$C. While AH13 previously found a mild tension between the SST data of MARGO and the SAT data of Bartlein et al. (2011), with the latter supporting a slightly colder analysis than the former, the TEA20 data appear to sit somewhere in the middle, reducing this hint of inconsistency. Thus, the slightly colder data of TEA versus MARGO does feed through into a colder overall result, but the effect is not substantial.

Using smaller observational errors also does not change the mean of the posterior, and if we try doing this then residuals we obtain are substantially larger than can be explained by observational error. For example if we use an error estimate of $1°$C for all data then the RMS value of the residuals for the posterior shrinks slightly to $1.9°$C, but this is of course now much larger than could be explained by the small observational error. In that experiment, the mean temperature change of the posterior is barely changed, with the global mean estimate becoming $-4.4\pm0.8°$C.

As a result of our tests, we consider our broadened posterior uncertainty range, compared to our earlier estimate, to be well founded and insensitive to reasonable choices. However, all of these results contrast strongly with the reconstruction of Tierney et al. (2020) where a posterior analysis estimate of $-6.1\pm0.2°$C was produced, an issue to which we return in the next section.

## 6   Discussion

Our new result of $-4.5\pm0.9°$C is slightly cooler than our previous analysis of AH13 where we obtained an estimate of $-4.0\pm0.4°$C. One reason for this is that the TEA20 SST data that we include here are slightly cooler than the MARGO data, with the mean difference between these two data sets being $0.3°$C at their co-located data points. However, the sensitivity tests presented in the previous subsections suggest that the different method and the new set of models also make small contributions to the additional cooling in the new result. Our new method is less susceptible to sampling noise in the model climatologies and

the result is visibly smoother than our previous analysis which had seemingly spurious anomalies for example in the Southern Ocean around (70°S, 20°W) and the north west Pacific at (60°N, 170°E).

A larger difference between the results is that the uncertainty is substantially higher in the new reconstruction, and this is due to different methodological assumptions. In the present work, the uncertainty is primarily derived from the spread of the ensemble, albeit this prior uncertainty is reduced in the neighbourhood of data points according to the Kalman Equations. In the AH13 reconstruction, the uncertainty was a heuristic estimate based on the results of the pattern scaling results when tested with synthetic (model-derived) data sets, and the data points all had global influence, in contrast to the localisation approach used here. Thus, in this new reconstruction, substantial regions of the Pacific ocean are only weakly constrained as few data points are available in the neighbourhood of these grid points. Our new approach is more in keeping with general practice for Bayesian estimation but does place a heavy burden on the model "ensemble of opportunity" as providing a reasonable representation of our uncertainty, especially when data are as sparse and uncertain as they are here. The area-weighted RMS difference between the SAT fields of our new result and that of AH13 over the spatial grid, is almost 2.4°C, or equivalently 1.2 standard deviations when normalised by the gridded standard deviations of the new result. Since these two reconstructions use overlapping sets of data and models, this suggests that the uncertainty on the new estimate is not unrealistically large and the lower uncertainty of the previous estimate may have been optimistic.

The greater difference between our reconstruction and that of TEA20 requires more detailed investigation. Using a similar Kalman filtering approach, TEA20 obtained a posterior estimate for global mean surface air temperature of $-6.1 \pm 0.2$°C, which is markedly colder than the result presented here, with a very much smaller uncertainty. While our mean estimate is very different to TEA20, our posterior ensemble does in fact contain one member which is similar to the TEA20 posterior mean. Figure 6 shows the latitudinally-averaged temperature profile for each of our posterior ensemble members, together with our ensemble mean. Substantial differences between ensemble members can be seen particularly in the Southern Ocean where the extent of sea ice may vary significantly between models. As explained in Section 4, each posterior ensemble member can be directly traced to the PMIP model simulation from which it was derived through the process of Kalman updating. The posterior sample originating from the CESM1-2 model is highlighted in green in this figure, with the other 18 ensemble members all shown as thin black lines. The figure also shows the posterior mean of TEA20, in purple. The similarity in latitudinal structure between these two climatologies is striking, and the overall cooling anomaly for this member of our ensemble is -6.0°C, very close to the posterior mean for TEA20 of -6.1°C. Furthermore, if we use the area-weighted RMS difference as a distance metric between fields of gridded SAT data, then the distance from the TEA20 posterior mean to this ensemble member is 1.8°C, which is lower than the distance to all other posterior ensemble members, and indeed considerably lower than the typical pairwise distance between most of our ensemble members themselves, which averages 2.6°C across the 171 possible pairs of fields. Despite the use of a largely different data set, a different prior covariance matrix, and numerous minor differences in the details of the algorithms, our posterior sample derived from CESM1-2 is notably similar to the TEA20 posterior mean, which demonstrates the importance of the underlying model field in interpolating the sparse and imprecise data points.

As an additional test, we performed the state estimation as described in Section 4, but instead of centering our prior on the EOF fit to the data, we instead centred it on the CESM1-2 simulation (i.e., set this field as the prior mean, with the

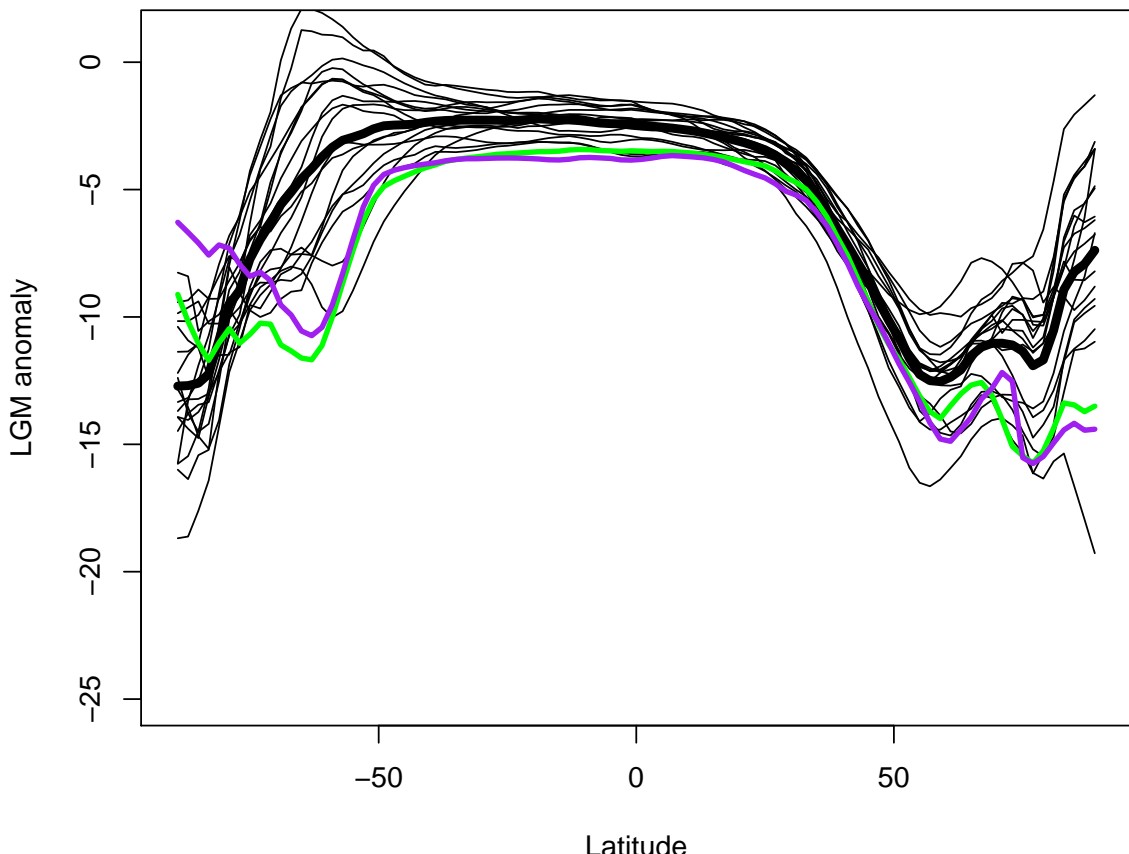

**Figure 6.** Latitudinal temperature anomalies. Thin black lines: ensemble members excluding CESM1-2 - derived member. Green line: CESM1-2 - derived member. Thick black line: posterior ensemble mean. Purple line: TEA20 posterior mean result

prior covariance unchanged). After performing the data assimilation, our posterior estimate for globally averaged temperature anomaly was $-6.2 \pm 0.8^\circ$ C, with the posterior mean field also being close to the TEA20 mean field on an area-weighted RMS difference basis.

Thus, our analysis suggests (in agreement with TEA20) that such a cold LGM state is plausible, but it also suggests that
5  considerably milder states are also compatible with the data. In fact while most (67%) of the TEA20 SAT reconstruction lies within the 2 standard deviation range of our estimate, less than 40% of our reconstruction lies within the 4 standard deviation range of the TEA20 result. That is to say, we consider the mean result of TEA20 to be reasonably compatible with the data and our own result, whereas conversely the TEA20 analysis strongly rejects most of our posterior range, and indeed also rejects the previous analysis of AH13.

10  While the prior is a personal choice that researchers may reasonably differ over, it seems doubtful to us that the prior of TEA20 is suitable for this problem. The single model prior of TEA20 does not consider any of the uncertainties in model

structure, forcing efficacies or or feedback strengths that contribute to our uncertainty regarding the LGM state. For example, all of the model instances in TEA20 will have the same equilibrium climate sensitivity. The reconstruction of TEA could not allow for the possibility of a significantly milder climate than was obtained, because this was excluded from their prior. This can be seen directly from Fig 2d in TEA where the prior 95% range is restricted to values for global mean temperature of colder than around -5.5C. This excludes not only the majority of the PMIP simulations — 15 from the 19 we used — but also the mean of the previous reconstruction of AH2013.

A multi-model ensemble, such as the one we are using, has been shown to represent uncertainty more realistically than any single model is able to do (Yokohata et al., 2011, 2013; Parsons et al., 2021). The reason for this is that the multi-model ensemble samples a range of structural differences and parameter values that lead to a broader spectrum of responses than can be obtained from a single model. One illustration of this is the broadly flat rank histogram of the data in our prior ensemble in Figure 3. This contrasts strongly with the rank histogram of the ensemble used by TEA20 which can be seen in Figure 2 of the Extended Data of that paper, and which shows a substantial under-dispersion such that much of the data they used lies outside the spread of the ensemble.

While our approach makes use of the diverse set of simulations generated by the multi-model ensemble that contributed to the PMIP experiments, we are limited to using the outputs that were generated by these experiments and cannot obtain diagnostics that are not already available or derivable from the variables that were saved. This limitation means we can only perform the analysis in temperature space, rather than in proxy space, which TEA20 were able to do using their proxy-enabled ocean model. Where possible, working in proxy space should be superior if there are sufficient proxy-enabled models and the modelling of the proxies is sufficiently skillful, as it can in principle account for ocean transport and mixing more realistically than a statistical calibration may do. However our approach has two important competing advantages to offset these limitations. The first of these is that we can use a wide range of proxy types without the need for development of climate models that include them as prognostic variables, so long as a calibration of these proxy data to local temperature is available. The second is that we can use a wide range of climate models, without needing to implement proxy models and integrate them ourselves. Both of these aspects allow us to consider a greater range of uncertainties and hopefully produce a more robust result.

## 7   Conclusions

We have presented a new reconstruction of the Last Glacial Maximum, with a global mean surface air temperature anomaly of $-4.5 \pm 0.9°C$ (with this uncertainty given at one standard deviation). This result is slightly cooler than the previous work of Annan and Hargreaves (2013), but markedly less cool than the recent result of Tierney et al. (2020). We have shown that the major reason for this discrepancy is that the prior of TEA20, based as it was on a single climate model which simulates a relatively cold LGM state, does not permit the broader and milder range of results that we have obtained.

While our approach is based on the well-established Ensemble Kalman Filter approach which has been widely used for a range of data assimilation tasks, we have shown that in this application, due to the limited availability of data, a biased prior

may strongly affect the result, and we have shown that bias-correcting the prior can be important for generating an accurate result.

The larger posterior range, and differences between this result and the previous analysis of AH13, does point to the importance of several major sources of uncertainty which limit the precision that can be achieved. Models generate very different climates, especially at high latitudes where the presence or absence of sea ice can result in widely varying air temperatures. Improvement of model simulations would, of course, help the creation of accurate climate reconstructions, but it is important that the range of models included in the PMIP ensembles represent all the main sources of uncertainty as realistically and comprehensively as practicable if they are to be used for this purpose. Substantial areas of the globe, such as much of the Pacific Ocean, are poorly served by proxy data, and our comparison of different data compilations suggests that proxy-based temperature estimates have substantial uncertainties. Better understanding and calibration of proxies would allow for a more precise result, and the widespread inclusion of forward modelling of proxies could also potentially help to reconcile these two sources of information.

## 8 Code and data availability

Code and data underpinning the analysis presented here is included as supplementary information.

## 9 Acknowledgements

We thank Jean Yves Peterschmitt and Masa Kageyama for creating and making available the monthly climatologies of TAS and TOS for the PMIP4 LGM models. This project was funded by the European Research Council (ERC) (Grant agreement No.770765) and the European Union's Horizon 2020 research and innovation program (Grant agreements No.820829 and No.101003470).

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
