# Peer review of "A new global surface temperature reconstruction for the Last Glacial Maximum"

_Climate of the Past, 2022_

## Referee Comment (RC1)

**Review of 'A new global climate reconstruction for the Last Glacial Maximum '**

The manuscript by Annan et al. is an interesting contribution to the literature on LGM temperature changes. The authors use a wide range of proxy-based reconstructions and available simulations. The resulting temperature fields thus can be seen as an aggregation of knowledge on LGM climate over the last few decades. Several important methodological advancements for the application of data assimilation in the paleoclimate context are presented in the manuscript, in particular the de-biasing of the prior ensemble and the a priori model selection to obtain more independent ensemble members. However, it is at times difficult to follow the presented results and explanations. Therefore, I recommend major revisions of the manuscript after which it should be a valuable resource for the community.

My major issues with the manuscript are as follows:

1. There are too few plots and the discrete color scales with often wide temperature steps make the current plots not very informative. The majority of SAT anomalies are between -2 and -8 K and should be represented by more than two different colors. The colors of the dots (proxy data) in Fig. 1 and 3 are difficult to identify. I was missing visualizations of

   - the difference between de-biased and not de-biased simulation ensemble (while the small difference in GMST is discussed in the text, a visualization of the spatially distributed difference is missing)

   - pairwise model similarity (the current discussion makes it impossible for the reader to understand how similar simulations needed to be such that only one of them was retained)

   - the validation results

   - the sensitivity tests

   - difference maps between the new reconstruction and discussed previous reconstructions (ideally including where differences are statistically significant)

2. A statement on data and code availability is missing. As the resulting fields will be a valuable community resource they need to be made available. To follow the choices made by the authors and compare them with previous approaches, code to reproduce the results should be made available.

3. The used error metrics in the validation and sensitivity sections are strongly focused on the posterior mean and the global mean temperature. As this is a Bayesian reconstruction of climate fields, a stronger focus on the spatial structures of the reconstruction (and how different choices discussed in the sensitivity tests influence it) and of the full posterior probability distribution would be more informative (e.g. maps of cross-validation results, coverage frequencies to study the meaningfulness of the posterior uncertainties,

probabilistic score functions). The included proxy data tends to be spatially clustered. Therefore, I wonder how much leave-one-out cross-validation is influenced by spatial autocorrelation and whether leaving our more data at once would be a better choice (e.g. h-block or leave-N-out cross-validation).

4. The abstract is too short and not very informative outside of the stated global mean temperature anomaly. I recommend to specify the used proxy data, simulations, and methodology. Actually describing the results from investigating the differences compared to Tierney et al. (2020) would be more informative for the interested readers than just writing "We discuss the reasons for this discrepancy".

5. The explanations on how statistical and data processing choices influence different features of the reconstruction are very valuable and should be a useful guide for future applications of data assimilation in paleoclimatology. However, they are often very general (e.g. p. 1 l. 2-3, 7; p. 2 l. 8-9; p. 3 l. 10, 26; p. 5 l. 16; p. 6 l. 12-13; p. 8 l. 9-10; p. 11 l. 12-14; p. 14 l. 4-5; p. 15 l. 5, 7-8, 19-20; see also the specific comments below) such that readers are forced to believe the authors and cannot trace the explanations in the results. Therefore, I recommend to specify these explanations.

Specific comments:

- Replacing "climate" by "temperature" in the title would be more precise.

- p. 1, l. 18-19 (and several subsequent instances): In which sense are the resulting fields 'physically consistent'? While the individual ensemble members are physically consistent, it is not clear to me how physically consistent the fields are after de-biasing and applying the ensemble Kalman filter.

- p. 2, l. 8-10: What are examples of absent localized features and suspected small-scale artifacts? Specifying the magnitude and spatial-scale of these features would be valuable to better interpret the results.

- p. 3, l. 10: What does "unusual ocean boundaries" mean? Has this been reported elsewhere? Given the numerous papers employing the PMIP3 ensemble, this finding of potential bugs in these simulations sounds relevant for others.

- p. 3, l. 26: What was used as cutoff values for RMS / pattern correlation? Giving more details seems necessary to interpret / reproduce the present details.

- p. 5, l. 16: What is the GMST anomaly of this outlier? What is the difference to the closest ensemble member? Given how much the GMSTs are used throughout the manuscript, plotting the GMST anomalies of the ensemble members (selected and removed ones) would be useful for the readers.

- p. 6, l. 12-13: How much of the variance is explained by the first four EOFs? How close is **q** to the final posterior mean?

- p. 7, l. 3: Cleator et al. (2020) augmented the Bartlein et al. (2011) data by Australian records from Prentice et al. (2017). These ones could be added to the dataset employed here. Given the new temperature reconstructions from ice cores that have been produced over the last decade, are the values from Schmittner et al. (2011) still up-to-date?

- p. 8, l. 13-15: Is the selection of TEA over MARGO based on the grid box averages or on the level of individual records? Given that TEA uses different archives than MARGO (inclusion of $\delta^{18}O$ in TEA, use of foraminiferal assemblages in MARGO) there might be some inconsistencies when the selection of MARGO over TWA is performed on the grid box level, in case there are systematic differences between different proxy types.

- p. 9-10: How is the specific implementation of the ensemble Kalman filter selected? How does it compare with the ensemble square-root Kalman filter employed in Tierney et al. (2020)? Are there previous examples of applying ensemble Kalman filters to multi-model ensembles? Are the assumption of it still satisfied for multi-model ensembles?

- p. 11, l. 12-14: A quantification of what "noticeably smoother" and "visually less structured" means would be very helpful.

- p. 14, l. 4-5: How is "no sign of over-fitting" measured? What would be considered over-fitting?

- p. 14, l. 18: It would help me if the rank histograms were shown and not just described.

- p. 14, l. 27-28: Which numbers of EOFs were tested? Assuming that the general cooling pattern is mostly contained in the first EOF (as they are not shown, I cannot determine that for sure), it would likely be more interesting to compare not just the GMST but the spatial patterns.

- p. 15, l. 5: How is the worsened fit to the data quantified?

- p. 17: Given the extensive sensitivity tests, are there general recommendations for the future usage of PMIP ensembles in climate field reconstructions and potentially for the design of future PMIP cycles? How could/should multi-model ensembles be designed for effective usage in climate field reconstructions?

**References**

Bartlein, P. J., Harrison, S. P., Brewer, S., Connor, S., Davis, B. A. S., Gajewski, K., Guiot, J., Harrison-Prentice, T. I., Henderson, A., Peyron, O., Prentice, I. C., Scholze, M., Seppä, H., Shuman, B., Sugita, S., Thompson, R. S., Viau, A. E., Williams, J., and Wu, H.: Pollen-based continental climate reconstructions at 6 and 21 ka: a global synthesis, Climate Dynamics, 37, 775–802, https://doi.org/10.1007/s00382-010-0904-1, 2011.

Cleator, S. F., Harrison, S. P., Nichols, N. K., Prentice, I. C., and Roulstone, I.: A new multivariable benchmark for Last Glacial Maximum climate simulations, Clim. Past, 16, 699–712, https://doi.org/10.5194/cp-16-699-2020, 2020.

Prentice, I., Cleator, S., Huang, Y., Harrison, S., and Roulstone, I.: Reconstructing ice-age palaeoclimates: Quantifying low-CO2 effects on plants, Global and Planetary Change, 149, 166–176, https://doi.org/10.1016/j.gloplacha.2016.12.012, 2017.

Schmittner, A., Urban, N. M., Shakun, J. D., Mahowald, N. M., Clark, P. U., Bartlein, P. J., Mix, A. C., and Rosell-Melé, A.: Climate Sensitivity Estimated from Temperature Reconstructions of the Last Glacial Maximum, Science, 334, 1385–1388, https://doi.org/10.1126/science.1203513, 2011.

Tierney, J. E., Zhu, J., King, J., Malevich, S. B., Hakim, G. J., and Poulsen, C. J.: Glacial cooling and climate sensitivity revisited, Nature, 584, 569–573, https://doi.org/10.1038/s41586-020-2617-x, 2020.

---

## Author Comment (AC2)

**Reply to Jessica Tierney**

Authors

10 May 2022

Thank you for the detailed and interesting questions and comments. To aid readability of our reply, specific changes proposed for our manuscript will be *highlighted in red*. Reviewer text quoted below is *highlighted in blue*.
* * *
One major change that should be highlighted at the outset is that *we will make our code and results available*. We probably should have done this initially, but wanted to make it a final version rather than having the risk of multiple versions being used. Therefore interested researchers will be able to check our result and further test it themselves in any way they wish. As so much of the review considers the contrast between our results (Annan et al, AEA subsequently) and those of Tierney et al (TEA), we will initially focus on this issue in some detail, before returning to the review comments on a point-by-point basis. We show below that a major cause of the differences between our results is in the respective choices of prior. The priors are substantially different and it is inevitable that this influences the results, especially in this situation where data are sparse and imprecise. Our prior does include a simulation from CESM1-2, the model that TEA used, and our posterior is sufficiently broad as to include much of the TEA range as well as milder values. Indeed as we show below, the ensemble member in our posterior corresponding to the CESM1-2 model is very close to the TEA reconstruction mean.

**1 Comparison with Tierney et al.**

*I think this is a useful contribution to the problem of reconstructing past climate fields and it is very interesting to see the outcome when a slightly different DA approach and multiple different model priors are used. However, the analyses in the paper as it is now don't identify \*why\* these new results are different than Tierney et al.*

While (as we replied earlier to Dan Lunt) our initial intention was not to investigate the analysis of TEA in detail, we do acknowledge that some further analysis is warranted, and interesting to the readers, and therefore have undertaken some further investigations which we describe in this Section. *We will*

*include in the paper extended discussion and comparison with TEA including a direct difference between the reconstructions (similar to Figure 1 attached here).*

**1.1 Description of differences between our AEA and TEA**

We agree that the Pacific plume in Fig 2a of TEA is not so anomalous, as you correctly stated in your review. Figure 1 displays the difference between the means of the AEA and TEA reconstructions. Two large scale discrepancies seem apparent. Firstly, there is a very strong anomaly in the Southern Ocean, where TEA is cooler by at least 4C (and up to 13C) over a substantial region. It seems possible that the strong cooling in TEA could be due to an increase in sea ice extent, as the SSTs in this area are much more similar between the two reconstructions, and there is a decoupling here between SST and SAT in the TEA reconstruction. However the important point is that the discrepancy exists, regardless of physical origin. The region shown in Figure 2 (defined as the region south of 46S where the TEA anomaly is at least -4C cooler than ours) represents about 5% of the global surface area, but contributes almost -0.4C to the difference in global mean temperature anomalies. PMIP models show a wide variety of results in this area as we shall illustrate shortly. There are very few data points in this region and SST proxy data cannot inform directly on SAT over sea ice as the ice insulates the air from the ocean surface.

The central latitudes of the Pacific basin as a whole (excluding the Southern Ocean areas described above, i.e. as defined in Figure 3) also has a substantial difference of around -2C in TEA relative to our reconstruction. Due to the large area this represents, this discrepancy contributes around -0.5C to the difference in our global means. While there are a few data points here, they are extremely sparse, and some show warming. There are a number of other regions showing differences in Figure 1, including the poles and Greenland, but the areas of these regions are too small for them individually to make really substantial contributions to the global mean temperature difference.

These two regions combined (covering a little more than a quarter of the globe) can explain a large majority of the difference between the global means of the reconstructions, which provides an account for the difference in our headline result, and also explains why both the TEA reconstruction, and our own, can agree reasonably with ice core data which are remote from these areas and hence uninformative regarding them.

**1.2 Explanation of differences between AEA and TEA**

In this subsection we show that the differences between the results are directly traceable to the difference in our priors, as we had surmised, and Figures 4 and 5 underpin our reasoning.

We used as our prior a large number of PMIP models that explore a wide range of uncertainty in feedbacks that determine their LGM states. Figure 4 shows the PMIP simulations that we use in our calculation (thin black lines, with

[Figure]

Figure 1: Difference between posterior means of TEA and this work.

the CESM1-2 simulation highlighted in red). We observe here that the CESM1-2 model exhibits the most extreme cooling in the Southern Ocean, and also shows the strongest cooling of the entire ensemble across the tropics especially south of the equator, which is dominated by the Pacific Ocean.

The posterior ensemble in the Ensemble Kalman Filter is generated by updating each member of the prior ensemble according to the Kalman Equation, and thus our posterior contains a member derived from applying this Kalman update to the CESM1-2 simulation. Figure 5 shows the latitudinal variation of our posterior ensemble members, together with the TEA reconstruction mean for comparison. The overall similarity between the latitudinal profile of our posterior sample based on CESM1-2 (red line), and that of the TEA reconstruction mean (purple line), is striking. This member of our ensemble appears more similar to the TEA reconstruction in structure and magnitude than does any other member of our ensemble.

[Figure]

Figure 2: Masking to show region of extreme cold in Southern Ocean. Difference in anomalies exceeds -4C in blue area, and this contributes almost -0.4C to the difference in global means between TEA and this work.

A simple numerical analysis supports this claim quantitatively. Table 1 summarises some statistics of our posterior ensemble. For each posterior ensemble member we list firstly the model it is sourced from in the prior, then the mean SAT anomaly of this posterior member, and lastly the area-weighted grid point RMS difference between this sample and the TEA posterior mean. The 12th member in the list, based on CESM1-2, is highly similar to the TEA posterior mean, having a SAT anomaly of -6C, and a grid point RMS difference from the TEA posterior mean of well under 2C, which is by some way the lowest value among all of our ensemble members. The ease with which this CESM1-2-derived sample can be identified by to its similarity to the TEA posterior mean is a strong indication of the influence of the prior, which persists despite substantial differences in prior covariance matrix, data set, and other methodological details. In both our calculation and that of TEA, the update of the

[Figure]

Figure 3: Masking to show region of strong cooling in Pacific Ocean. Difference between LGM anomalies averages in excess of -2C in this area, and this region contributes -0.5C to the total difference in global means between TEA and this work.

CESM1-2 model leaves us with something that is still recognisably CESM1-2.

We therefore see that our posterior ensemble contains a member which is close to the TEA ensemble mean, and as a result, our posterior result more-or-less includes that of TEA. Specifically, the TEA reconstruction's global mean temperature estimate is within our 2sd range, as is most (67%) of the mean spatial map on an area-weighted gridpoint basis. Conversely, however, our posterior global mean temperature estimate of -4.5C is far outside of the TEA posterior 95% range, and a large majority of our spatial map (over 60% of the globe) is outside the 4sd range of the TEA spatial estimate. This is despite our reconstruction showing a high level of agreement with not only the data used in TEA, but the other diverse and independent data compilations that we used.

| Model name | Global mean SAT anomaly | RMSD vs TEA posterior mean |
|---|---|---|
| CCSM | -4.18 | 3.05 |
| CNRM | -3.13 | 4.75 |
| FGOALS | -5.16 | 2.87 |
| HadCM3 | -4.62 | 3.36 |
| IPSL | -3.59 | 4.05 |
| ECHAM | -4.96 | 2.9 |
| FGOALS | -4.25 | 3.58 |
| GISS-E2 | -4.48 | 3.47 |
| MPI | -4.03 | 3.5 |
| MRI | -4.34 | 2.87 |
| CCSM4 | -4.81 | 2.52 |
| **CESM1-2** | **-6.02** | **1.88** |
| HadCM3-GLAC1D | -5.73 | 2.24 |
| HadCM3-PMIP3 | -6.18 | 2.9 |
| INM-CM4-8 | -3.56 | 4.3 |
| IPSLCM5A2 | -4.54 | 3.48 |
| MIROC-ES2L | -3.74 | 3.56 |
| MPI-ESM1-2 | -3.79 | 3.43 |
| iLOVECLIM1-1-1-ICE-6G-C | -3.71 | 3.45 |

Table 1: Summary of posterior ensemble members, showing mean temperature anomaly at LGM and also RMS difference of spatial map with that of TEA posterior mean

[Figure]

**Latitudinal variation of initial LGM anomalies of PMIP ensemble**

Thick black line: ensemble mean

Red line: ensemble member based on CESM1−2

Thin black lines: ensemble members excluding CESM1−2

Figure 4: Latitudinal summary of our prior ensemble. CESM1-2 simulation is highlighted in red, with the other ensemble members shown as thin black lines. Thick black line is ensemble mean.

**1.3 New sensitivity test - Recentering the prior**

As a corollary of this analysis, if we recentre our entire ensemble on the CESM1-2 model output, rather than centering it on the data as described in the manuscript, then we can anticipate that our posterior mean would be very similar to that of TEA. In fact when we perform this experiment, our posterior global mean temperature anomaly becomes -6.4C and our mean field is again within 2C of TEA on a gridpoint RMS basis. Thus, the difference in the reconstruction means of TEA and AEA can be directly traced to the difference in prior means. *We will report this experiment in our manuscript.*

**1.4    Comments on choice of prior**

While the prior is a personal choice that researchers may reasonably differ over, it seems doubtful to us that the prior of TEA is suitable for this problem. The prior of TEA does not consider any of the uncertainties in model structure or feedback strengths that contribute to our uncertainty regarding the LGM state. As a consequence, all of the model instances in TEA will have the same equilibrium climate sensitivity. Sampling internal variability does little to mitigate these limitations. The overconfidence of the prior in TEA is apparent from the rank histograms in the supplementary data; the reconstruction of TEA could not allow for the possibility of a significantly milder climate than was obtained, because this was excluded from the prior. This can be seen directly from Fig 2d in TEA where the prior 95% range is restricted to values for global mean temperature of colder than -5.6C. This excludes not only the majority of the PMIP simulations — 15 from the 19 we used — but also the mean of the previous reconstruction of Annan and Hargreaves 2013 (AH2013). While a Gaussian has no absolute boundaries, many of these values are outside even the $\pm 3SD$ (99.7% probability) range of the TEA prior. It was therefore inevitable that the TEA posterior would also be restricted to a much colder result than AH2013 and would exclude the values generated by many of the PMIP simulations, regardless of the data.

**1.5    Summary of comparison with Tierney et al.**

Based on this analysis, we do not think there is further reason to perform the many additional tests and calculations suggested in order to explore why the AEA and TEA differ. We agree that the data are not inconsistent with a cold ($\simeq -6C$) LGM, at least at one extreme of our posterior range. Indeed we generate a posterior sample that is very close to the TEA reconstruction mean, directly through applying the Kalman update to the member of our prior ensemble generated by the model that was used in TEA. We have however shown that the data are also consistent with much less extreme cooling than TEA supported, with substantial uncertainty appearing to arise from the different ways that models extrapolate into poorly observed areas. TEA excluded such possibilities *a priori* and therefore cannot attribute this aspect of their result to the new data. Indeed a direct comparison of the gridded proxy data of TEA with that of MARGO at coincident locations suggests that it represents rather similar conditions, albeit marginally cooler, and this is also broadly consistent with the pollen data of Bartlein et al. The TEA prior is very narrow and centred on far colder values than the AEA prior. We do not believe there is any further discrepancy to be explained between our results, beyond what would be reasonably expected as a direct result of these two factors.

**2 Specific comments**

We now address the specific points in your review.

*Osman et al., (2021) which finds an even cooler LGM (-7C) with a more limited proxy network yet much wider priors (this follow-up paper isn't discussed here, but should be!)*

We didn't discuss Osman et al as it is not primarily a reconstruction of the LGM and itself makes the point that it uses fewer data points at each interval in time, therefore cannot be expected to be as accurate a result for the LGM specifically. It seems inevitable that it also suffers from the problem that TEA has of relying on the internal variability of a single model to generate an ensemble, which by construction does not include uncertainties in climate system feedbacks. If the topic of our manuscript were a multi-model data assimilation reconstruction of the deglaciation, it would be more relevant to include comparison with Osman et al.

*Conversely, it seems like the relatively warm priors here are influencing the result also, given that the adjusted prior mean is -4.9 p/m 1.1C.*

The prior is centred on the data via the initial translation step and thus is neither relatively warm nor cool.

*The authors also find that if they inflate the prior mean cooling to -9.7C, they get a substantially cooler posterior (Page 14), unless they do their "translation procedure", which re-centers the prior on the data a priori. But then, isn't the result basically dependent on pre-centering the prior on the data – which means you presuppose what level of cooling the data show? I think this could be problematic. The data might show a mean cooling of say -4.5C, but the data don't sample the coolest places on Earth (like the Laurentide) so this is almost certainly an underestimate. To what extent does the translation step bake-in the posterior result?*

Since we do the recentering on a spatial basis using gridded data, the location of data points will not in itself bias the result if the spatial patterns of the models are reasonable. Thus cold unobserved places like the Laurentide ice sheet do not affect the translation.

*In Tierney20, we calculated a proxy-only global mean cooling to compare with the DA and found it has a median of -5.5C. This was based only on SST, so reasonably could be too warm since it excludes land data. So first order, the -4.5C result here just does not seem cold enough. Granted the -5.5C we calculated has to assume a scaling b/t SST and SAT (we followed Snyder et al, 2016 and drew this scaling from PMIP). Nevertheless you could calculate a proxy-only value here for comparison, and also compute a proxy-only global SST (which doesn't have to be scaled). Can you calculate proxy-only global SST change and*

*compare to the DA results (as we did in Fig. 2 of Tierney20)? We got -2.9C with the TEA network. I'm just curious whether the DA proxy posterior comes out similar to this or warmer.*

We emphasise that our posterior range includes -5.5C (indeed this value is only just over 1sd from our mean) and such a result is therefore not particularly unlikely. We do not think there is any substantial discrepancy to explain here. The simple mean of the 405 data points we used is -3.2C, which is certainly an underestimate of the global mean temperature change as the data are dominated by SST proxies and are not missing at random. Performing the fit at the correct locations accounts for this, on the assumption that ensemble of models contains reasonable spatial patterns.

*In Tierney20, for sure our prior was a bit too tight than we liked which we discussed in the paper and is evident in the U-shape of the rank histograms. However, interestingly, there was no strong mean bias in the LGM rank histogram (but there was a slight mean bias for the Late Holocene) which suggested to us that the LGM prior was not too cold; otherwise we would expect to see a skewed shape in the rank histogram. This is different than the rank histograms shown here which are much more flat. I'm not sure how to reconcile this. Our rank histograms are based on withheld validation data only (not sure if that is the case here or not). They were also calculated in proxy space, taking advantage of the forward modeling in our DA technique. So how to compare those to these?*

The direct reconciliation of the U-shaped rank histogram of TEA versus the flatter one of AEA is that the TEA prior is strongly overconfident, which is as expected given the use of a single model. Our prior has a much broader range of model responses, due to including structural uncertainties (as represented by the PMIP models) in its construction. This result is consistent with several papers we have published over the past decade (eg Yokohata et al, 2012 and 2013), and also an independent analysis that was published more recently (Parsons et al, 2021).

We agree that your rank histograms do not show a strong bias, and the same is true of ours. While this might on the face of it seem inconsistent, the data are not missing at random, as we noted in AH2013, which makes the extrapolation into data voids a highly model dependent process. Therefore, it is necessary to consider the full range of plausible model results rather than focussing on just one model, especially given that the CESM1-2 model appears to be such an outlier for the data-poor Southern Ocean and Pacific mid-latitudes. It is also worth mentioning that the rank histogram is not a very comprehensive test, and while failing to obtain a reasonably flat rank histogram is certainly indicative of a problem, it is also quite possible for a poor ensemble to pass this test satisfactorily. A flat rank histogram is a necessary but not sufficient condition.

*ERSSTv5 is used here as a preindustrial baseline instead of the Late Holocene*

*values. What happens if you use the Late Holocene SSTs in TEA instead? Our reasoning for using Late Holocene SST was that in some locations there is a strong bias in the proxies (relative to climatological SST) so subtracting LH SST would correct for this. This said, LH SST estimates from the TEA network might suffer from non-modern coretops. In Tierney20, we did not screen the data for age control, so some of the coretops might have pre-modern (likely colder) proxy information. We speculated that this is one reason why Osman et al LGM cooling is larger. In that study, we only used proxy data with good age control so the Late Holocene data are for sure Late(est) Holocene. Perhaps it is worth running your DA using the screened proxy network of Osman et al. alone?*

As described in the text, we found a slightly better agreement between MARGO and TEA when we used their absolute values and a common pre-industrial baseline, versus the published anomalies for both (which use different baselines). However it is worth emphasising that the difference is minor. Since our method is based entirely on anomalies, using the core-top data of TEA would restrict us to the subset of gridpoints where both core-top and LGM are co-located. We accept there are arguments in favour of using core-top in that biases may cancel, but on the other hand, it requires another set of imperfect measurements and calibration. *We will publish our code so anyone interested in performing a wider set of calculations will be free to do so.* However, we think our choices are reasonable. If there were major internal inconsistencies between the respective anomaly and absolute data sets of either MARGO or TEA, this would be problematic, but we don't see any evidence for this.

*Working in temperature vs. proxy space. I suspect this could produce some of the differences between the result here and Tierney et al (and Osman et al). For one, in these latter studies the forward modeling allows us to consider the seasonality of the proxies and assimilate them during different production seasons. In contrast, it seems that here all of the proxies are treated as annual mean T (?). What happens if, instead, you consider the seasonality and assimilate the proxies in their respective seasons (as we did)?*

We performed seasonal reconstructions in Annan and Hargreaves, Quaternary Science Reviews 107 (2015) using MARGO and Bartlein et al data and found only modest differences from our annual mean calculation. *We will cite this paper and mention the seasonal aspect.* We do agree that working in proxy space should be intrinsically superior if the proxy models are good enough. Of course our results depend on the accuracy of all of the gridded data sets we use, a point that we make in the manuscript. We have however already shown that the choice of prior can explain the differences between our results. If the proxy assimilation produces significantly different values from the calibrated estimates at those grid points then that would be interesting, but we suggest that is a question for the respective scientists to address, not us.

*Page 5, Line 20: I agree that CESM2 is an outlier and probably needs to*

*be removed to satisfy the condition of normality, but it is noteworthy that the resulting ensemble mean is 4.5 p/m 1.1, which is rather tight.*

Please note that the uncertainty quoted here is 1sd and thus our prior uncertainty on the global mean is rather more than double that used by TEA. We therefore do not agree that it is rather tight, but we do report on tests where it was inflated.

*The authors can test the theory that the CESM priors are the main source of the difference by applying their approach to a CESM-only ensemble.*

We have shown above that centering our prior on the CESM1-2 simulation provides a posterior mean very close to that of TEA. Further tests of the priors don't seem necessary.

*Finally, validation. One of the strongest pieces of evidence that the solutions of Tierney20 and Osman21 are robust is the fact that we excellent independent validation with the ice core data. If LGM cooling were only -4.5C globally, I doubt we would get as good of validation. In contrast, there is no external validation in this study*

The reason that there is no additional data other than that we used, is that we used all the data that were readily available so as to generate as high a quality result as possible. While we could have presented a main result that only used some of the data, it would be less well constrained and thus we do not consider this criticism valid. The result we obtain is essentially unchanged for all of the sets of 404 data points anyway, and the left out data provides the validation in these cases.

*Finally, Seltzer et al. (2021) ...*

There are many interesting analyses of a small number of data points giving a range of results often focussed on particular regions. However, we are not experts in the interpretation of data from proxies. Our aim with this work is rather to generate a comprehensive global analysis using the available large and widely accepted compilations of data, and publicly available model outputs, in order to estimate what these information sources combined can tell us about LGM temperatures.

*Page 8, line 15: The TEA database includes all of the data from MARGO that are d18O, Mg/Ca, and UK37, so if you are adding MARGO estimates double check that it is only the assemblage data, which we did not use. Otherwise the same data might be used more than once when you combine the datasets.*

Where data are co-located, we use the TEA data set in preference to MARGO, rather than averaging the two together. Thus there is no double-counting.

*Page 10, line 10: Can you clarify here whether you are doing serial updates (one observation at a time) or a joint update (all observations together)? Tierney et al. and Osman et al. are joint updates. There can be some differences*

*b/t using serial vs. joint although they are usually small.*

The calculation is joint.

*Page 11, line 14: The authors take issue with the cold N. Pacific gyre signal in T'20, but they also show this feature in their reconstruction with a cooling of -4 to -8. Indeed, most PMIP models indicate a strong cooling here and the proxy data also suggest this. In T'20 it is indeed larger (-6 to -10), but I don't think we can say which magnitude is more accurate given the uncertainties in the proxies, and also the lack of proxy data from the center or eastern side of the gyre. Note that Gray et al. 2020 (PaleoPaleo) document gyre cooling of about -5C on average, but deeper cooling is seen in some of the raw Mg/Ca and UK37 data. It seems to me that it's hard to argue what the exact magnitude is. I would either drop this sentence or introduce more nuance here.*

Agree as mentioned previously. *We will refer instead to the Southern Ocean and Pacific more generally, and intend to display the direct comparison of our gridded mean results in the form of Figure 1 here.*

*Page 12: Rank histograms. Are these calculated on the withheld validation data (they should be)? Please clarify.*

We did not combine the test of biased priors, together with the leave one out analysis. Thus the question does not arise here.

*Page 17, Line 9: I would not go so far as to argue that using calculated SST/SAT from the proxies is better than forward modeling, i.e. working in proxy space. This \*might\* be true for proxies that are univariate, but it is definitely not true for proxies that are not, which includes d18O, Mg/Ca, foram transfer functions, and pollen. To translate these to temperature, one has to make assumptions about the other environmental influences on the these proxies (i.e. pH, salinity, or pollen, moisture balance) that are going to be imperfect compared to the forward modeling scenario in which multiple environmental parameters can be accounted for. Unless you can prove otherwise, using derived temperatures will be inferior to working in proxy space where inversion is not needed and you can make fewer static assumptions.*

We agree and explicitly state that forward modelling of proxies, where available, has the potential to be superior to using SST/SAT from a statistical calibration. *We will reword this to be clearer.*

*Page 11, line 16: With this many data points, I think leaving out a percentage of the proxy data, as was done in Tierney et al. and Osman et al., is a more robust test than leave-one- out. I suggest withholding 20%-25% of the data.*

Leave one out allows a comprehensive test of predictive ability for the scheme as published. The main reason for leaving out a larger random set would be if computational costs precluded testing each data point in isolation, which is not the case here. We did however also test the omission of various combinations of TEA, MARGO and Bartlein et al, which is a far stiffer challenge than using

random subsets as it tests the possibility that these data sets have different characteristics such as coherent biases due to the different approaches of the different groups of researchers. However, as described in the text, we found only modest differences.

[Figure]

Figure 5: Latitudinal summary of our posterior ensemble. Sample based on CESM1-2 simulation is highlighted in red, with the other ensemble members shown as thin black lines. Thick black line is ensemble mean. Purple line indicates TEA posterior mean.

---

## Author Comment (AC3)

**Reply to Referee #1**

**Authors**

**May 2022**

Thank you for the detailed and interesting questions and comments. To aid readability of our reply, specific changes proposed for our manuscript will be *highlighted in red*. Reviewer text quoted below is *highlighted in blue*.
* * *
One major change in response to major point 2 that should be highlighted at the outset is that certainly *we will make our code and results available*. Therefore interested researchers will be able to check our result and further test it with their own calculations and diagnostics in any way they wish. This should also help with the various comments about the details of our approach, the interpretation of results, and possible further tests.

*1. There are too few plots and the discrete color scales with often wide temperature steps make the current plots not very informative. [...]*

*The number of plots will increase significantly in our revision*, with some specific additions mentioned here and also in our response to the other reviewer. However, we believe that making our code and output available should satisfy most requests more effectively than producing a large number of figures that we believe will be of little interest. Perhaps we have misunderstood your requests but there would be 378 figures of pairwise model differences and 405 plots of leave-one-out validation tests. Both of these numbers would double if we included both SST and SAT. We are thinking that this is a rare case of a few words being worth a thousand pictures.

As for the colour scheme, it was originally used in AH2013 and was based on the draft of IPCC AR4 which was circulating at that time. We actually thought our nonlinear scale with 1 degree bins close to 0C added useful clarity compared to the 4 degree bins (+2 – -2C, -2 – -6C etc) that the IPCC used. 11 colour bins doesn't seem particularly out of line with other similar papers, eg Fig 2d of Tierney et al and Figure 4e of Cleator et al. We agree that the figures are challenging to interpret precisely and spent some time working on them but we are not experts in graphical design and since numerical output will be provided, the figures are only needed as a summary for those who do not require high precision. We hope that supplying the code and data will satisfy most of your criticisms as this will enable all researchers to perform further calculations and re-plot the data in their preferred style.

*2. A statement on data and code availability is missing.*

Yes, we apologise about this and *will include a statement to the effect that data and code will be available as supplementary information.* We were originally planning to make these available at the revision stage, but clearly we should have done so for the initial submission.

*3. The used error metrics in the validation and sensitivity sections are strongly focused on the posterior mean and the global mean temperature. As this is a Bayesian reconstruction of climate fields, a stronger focus on the spatial structures of the reconstruction (and how different choices discussed in the sensitivity tests influence it) and of the full posterior probability distribution would be more informative (e.g. maps of cross-validation results, coverage frequencies to study the meaningfulness of the posterior uncertainties, probabilistic score functions). The included proxy data tends to be spatially clustered. Therefore, I wonder how much leave-one-out cross-validation is influenced by spatial autocorrelation and whether leaving our more data at once would be a better choice (e.g. h-block or leave-N-out cross-validation).*

We have already undertaken 'extensive sensitivity tests' (in your own words below). We have specifically tested omitting large subsets based on TEA, MARGO and Bartlien et al which we consider to be a particularly strong challenge as these different researchers have collated and calibrated their data in different ways. *We will add a figure showing a rank histogram of the leave-one-out validation results.*

*4 The abstract is too short and not very informative outside of the stated global mean temperature anomaly. I recommend to specify the used proxy data, simulations, and methodology. Actually describing the results from investigating the differences compared to Tierney et al. (2020) would be more informative for the interested readers than just writing "We discuss the reasons for this discrepancy".*

*We will improve and extend the abstract, including a more explicit outline of our method and comparison of our results with other work.*

*5. The explanations on how statistical and data processing choices influence different features of the reconstruction are very valuable and should be a useful guide for future applications of data assimilation in paleoclimatology. However, they are often very general(e.g. p. 1l. 2-3,7;p. 2l. 8-9;p. 3l. 10,26;p. 5l. 16;p. 6l. 12-13;p. 8 l. 9-10; p. 11 l. 12-14; p. 14 l. 4-5; p. 15 l. 5, 7-8, 19-20; see also the specific comments below) such that readers are forced to believe the authors and cannot trace the explanations in the results. Therefore, I recommend to specify these explanations.*

*We will improve the text, and code will also be made available which clarifies the details.* Our results are rather insensitive to the choice of various parameters in the scheme and the final choices made are somewhat subjective.

Specific comments

*Replacing "climate" by "temperature" in the title would be more precise.*
*Agree.*

*p. 2, l. 8-10: What are examples of absent localized features and suspected small-scale artifacts? Specifying the magnitude and spatial-scale of these features would be valuable to better interpret the results.*
*We will describe in more detail along with the comparison to previous work.*
For example, there was (what appeared to be) a notable artefact in the Southern Ocean of the AH2013 reconstruction, remote from data.

*p. 1, l. 18-19 (and several subsequent instances): In which sense are the resulting fields 'physically consistent'? While the individual ensemble members are physically consistent, it is not clear to me how physically consistent the fields are after de-biasing and applying the ensemble Kalman filter.*
Consistent in the linear sense, an approximation which is intrinsic to the Kalman Filter (along with the assumption of gaussianity). The contrast we are drawing is with simpler assimilation schemes (and ordinary statistical interpolation approaches) which don't achieve this. *We will clarifiy the wording.*

*p. 3, l. 10: What does "unusual ocean boundaries" mean? Has this been reported elsewhere? Given the numerous papers employing the PMIP3 ensemble, this finding of potential bugs in these simulations sounds relevant for others.*
We doubt that this is a bug in the implementation of the PMIP protocol or the model simulation itself. What we have is model output downloaded from ESGF on an ocean grid that does not interpolate well onto our 5 degree grid, losing a lot of coastal area in the process. Researchers who are not using similar gridded (primarily coastal) data probably won't notice a problem. It may be that more sophisticated processing (we used standard routines in CDO) could resolve the issue but, as we already had alternative models from these centres, this did not seem worth pursuing.

*p. 3, l. 26: What was used as cutoff values for RMS / pattern correlation? Giving more details seems necessary to interpret / reproduce the present details.*

We did not use a fixed cutoff, but selected the models as described. We approached this decision (and many others regarding the detailed implementation of the algorithm) as a fundamentally subjective one, with tests and calculations used as a guide rather than a strict but arbitrary rule. Making code available will allow others to test different choices if they wish.

*What is the GMST anomaly of this outlier? What is the difference to the*

*closest ensemble member? Given how much the GMSTs are used throughout the manuscript, plotting the GMST anomalies of the ensemble members (selected and removed ones) would be useful for the readers.*

*We will include histograms of the GMST of the 28 and 19 member ensembles, and the posterior.* That model has an anomaly of about -11C, with the others ranging from around -3C to -7C.

*How much of the variance is explained by the first four EOFs? How close is $q$ to the final posterior mean?*

*Values will be included.*

*Cleator et al. (2020) augmented the Bartlein et al. (2011) data by Australian records from Prentice et al. (2017). These ones could be added to the dataset employed here. Given the new temperature reconstructions from ice cores that have been produced over the last decade, are the values from Schmittner et al. (2011) still up-to-date?*

We are reluctant to go down the route of chasing up small updates and changes, especially as we are not expert in such data analyses and compilation ourselves. While we do seek to use recent, large, and openly available datasets, such that our results are credible, our main focus is on the assimilation of data and models, which we believe is an important and sometimes under-recognised skill in itself. We do hope that others with greater expertise in climate proxy data may be able to implement our code incorporating their own data.

*Is the selection of TEA over MARGO based on the grid box averages or on the level of individual records? Given that TEA uses different archives than MARGO (inclusion of d18O in TEA, use of foraminiferal assemblages in MARGO) there might be some inconsistencies when the selection of MARGO over TWA is performed on the grid box level, in case there are systematic differences between different proxy types.*

All of our work is at the grid box level, and indeed these authors quite possibly (probably) interpreted the same data in different ways. Where the data sets coincided, we used TEA in preference to MARGO, which avoids any possibility of double counting.

Our tests of the TEA and MARGO data sets, both in the assimilation and also via a direct comparison at the gridpoint level was specifically aimed at checking for inconsistencies. We did find that TEA represents very slightly cooler conditions, but the discrepancy appears small. However the RMS difference between the data sets was significant, a point of evidence that we use and discuss in our estimation of observational uncertainties.

*How is the specific implementation of the ensemble Kalman filter selected? How does it compare with the ensemble square-root Kalman filter employed in Tierney et al. (2020)? Are there previous examples of applying ensemble Kalman filters to multi-model ensembles? Are the assumption of it still satisfied*

We are not sure if the EnKF has been applied to multi-model ensembles, but it certainly has with perturbed physics ensembles. The linear and gaussian approximations are not strictly true for any complex nonlinear situation, including numerical weather prediction where this approach has a long and successful history.

Our specific choice of algorithm was made for convenience and familiarity. TEA contains references for their algorithm. It is a slightly different implementation of the same fundamental equations (i.e. the Kalman equations).

*p. 14, l. 4-5: How is "no sign of over-fitting" measured? What would be considered over-fitting?*

Over-fitting was checked by looking for a worsening of the fit to withheld data as the number of EOFs was increased (and this did not happen). See AH2013 for previous analysis, though this used the model fields for the fitting rather than an EOF decomposition. The choice of number of EOFs to use is subjective, as is the case for a number of other parametric choices we made. Since we do not believe that there is an objectively correct way to perform the analysis, we tested here and elsewhere that the results were robust to the choices made.

*It would help me if the rank histograms were shown and not just described. We will include these rank histograms.*

*p. 14, l. 27-28: Which numbers of EOFs were tested? Assuming that the general cooling pattern is mostly contained in the first EOF (as they are not shown, I cannot determine that for sure), it would likely be more interesting to compare not just the GMST but the spatial patterns.*

All values from 1 to 8, and also the full set. With the uncentred approach, the first EOF is close to the ensemble mean and the subsequent ones represent variability at increasingly finer scales. There is no obvious physical interpretation however.

*p. 15, l. 5: How is the worsened fit to the data quantified?*
By the RMS of the residuals.

*Given the extensive sensitivity tests, are there general recommendations for the future usage of PMIP ensembles in climate field reconstructions and potentially for the design of future PMIP cycles? How could/should multi-model ensembles be designed for effective usage in climate field reconstructions?*

*We will add some discussion about this.* It has been a frequent topic of discussion within PMIP meetings but may be useful to a wider audience. Design of experiments needs to sample the relevant uncertainties, which in the case of paleoclimate will normally include forcings and other boundary conditions, as

well as uncertainties in physics including feedbacks which are sampled across the ensemble of models.